

# An uncertainty methodology for solar occultation flux measurements: ammonia emissions from agriculture

Johan Mellqvist[1], Nathalia T. Vechi[1,2], Charlotte Scheutz[2], Marc Durif[4], Francois Gautier[4], John Johansson[3], Jerker Samuelsson[3], Brian Offerle[3] and Samuel Brohede[3]

[1]Department of Space, Earth and Environment, Chalmers University of Technology, Göteborg, Sweden
[2]Department of Environmental Engineering, Technical University of Denmark, Lyngby, Denmark
[3]Fluxsense AB, Göteborg, Sweden
[4]INERIS, Verneuil-en-Halatte, France
*Correspondence to*: Johan Mellqvist (johan.mellqvist@chalmers.se)

**Abstract.** Ammonia ($NH_3$) emissions can negatively affect ecosystems and human health, so they should be monitored and mitigated. This study introduces a novel methodology for evaluating uncertainties in $NH_3$ emissions measurements using the Solar Occultation Flux (SOF) method. The reactive nature of $NH_3$ makes its measurement challenging, but SOF offers a reliable open-path passive method, utilizing solar spectrum data, thereby avoiding gas adsorption within the instrument. To compute $NH_3$ gas fluxes, horizontal and vertical wind speed profiles, as well as plume height estimates, and spatially resolved column measurements are integrated. A unique aspect of this work is the first-time description of plume height estimations derived from ground and column $NH_3$ concentration measurements aimed at uncertainty reduction. Initial validation tests indicated measurement errors between -31 % and +14 % on average, which was slightly larger than the estimated expanded uncertainty ranging from ±12 % to ±17 %. Application of the methodology to assess emission rates from farms of various sizes showed uncertainties between ±21 % and ±37 %, generally influenced by systematic wind uncertainties and random errors. The method demonstrates the capacity to measure $NH_3$ emissions from both small (~1 kg h⁻¹) and large (~100 kg h⁻¹) sources in high-density farming areas. Generally, the SOF method provided an expanded uncertainty below 30 % in measuring $NH_3$ emissions from livestock production, which could be further improved by adhering to best application practices. The findings also have implications when using SOF to measure other gaseous species and in other applications.

## 1. Introduction

Agriculture is the primary source of ammonia ($NH_3$) emissions, accounting for around 85 % of total discharges globally (EDGAR database, 2023) – a figure that has increased since pre-industrial times due to growing food demand (Galloway et al., 2003). Among the different agricultural sources, livestock production releases $NH_3$ due to animal urine and faeces decomposition. $NH_3$ is a precursor of atmospheric fine particulate matter ($PM_{2.5}$), eutrophication and an indirect greenhouse gas (GHG). $PM_{2.5}$ is associated with lung diseases, and $NH_3$ accounts for approximately 30 and 50 % of $PM_{2.5}$ in the US and



Europe, respectively (Wyer et al., 2022). The atmospheric lifetime of $NH_3$ ranges from hours to days, as it can either react in

the atmosphere forming $PM_{2.5}$ or be retained in the ground due to dry or wet deposition. The complex emissions, reactions and deposition mechanisms of $NH_3$ hinder our understanding of these emission sources and associated dynamics (Hristov et al., 2011), so there is a need to monitor $NH_3$ emissions and atmospheric concentrations (Wyer et al., 2022). Knowledge gaps still need to be filled regarding $NH_3$ emission dynamics, which is reflected in the large discrepancies between modelled $NH_3$ and measured emissions (Lonsdale et al., 2017). A recent study on $NH_3$ emission hotspots using satellite data indicated that two-

thirds of high emission sources are underestimated by at least one order of magnitude (Van Damme et al., 2018). Furthermore, in Europe, $NH_3$ emissions are regulated under EU law (NEC 2016/2284) by reporting, monitoring and limiting emissions under certain thresholds (Wyer et al., 2022), which requires the development of emissions reduction technologies and reliable quantification techniques.

Consequently, $NH_3$ has gained attention over the last few decades, thus increasing the development of instruments and models

used to study its emission sources. Moreover, with improvements in infrared lasers, spectroscopy-based instruments have emerged, such as FTIRs (Fourier Transform infrared spectrometers), cavity ring-down spectrometers (CSDRs) and quantum cascade laser absorption spectrometers (QCLASs) (Twigg et al., 2022). $NH_3$ concentrations are challenging to quantify due to its strong reactivity, which makes the gas molecule adhere to surfaces and requires that close-path instruments and inlets are coated or heated to decrease the response delay (Zhu et al., 2015b). A study using 13 instruments highlighted the importance

of its setup, inlet design and operation (flow rate and filter status), as these factors can affect measurement performance (Twigg et al., 2022).

Furthermore, measurements can be taken from mobile (Eilerman et al., 2016; Golston et al., 2020; Miller et al., 2015), stationary (Sun et al., 2015a) or airborne platform (Guo et al., 2021; Miller et al., 2015; Sun et al., 2015b). Mobile platforms can resolve local scales very well (Golston et al., 2020), even though they are limited by road availability. Furthermore,

Lassman et al. (2020) found that a surface-based platform can underestimate $NH_3$ emissions by a factor of 1.5 because concentrations near the surface might be depleted due to gas deposition. In addition, in recent years, satellite column retrievals have complemented information on $NH_3$ emissions from large-scale sources. These platforms have extensive spatial coverage but suffer from high emission uncertainties and poor spatial and temporal resolution.

The Solar occultation flux (SOF) has been used for years in the quantification of alkenes, VOCs and industrial $NH_3$ (Baidar et

al., 2016; Johansson et al., 2014; Mellqvist et al., 2007, 2010) and has been recently used to measure agricultural $NH_3$ emission sources (Kille et al., 2017, Vechi et al., 2023). The SOF technique measures spatially distributed slant columns (g m$^{-2}$), which can be converted to emission rates using additional information about wind speed and direction.

This approach can complement in-situ and satellite measurements, effectively bridging these two techniques (Guo et al., 2021). The uncertainty with this technique has been briefly discussed before for VOCs (Johansson et al., 2013), alkenes (Mellqvist et

al., 2010) and $NH_3$ (Kille et al., 2017). Herein, our aim is to further explore the error analysis with a comprehensive measurement uncertainty methodology and a comparison to validation experiments. Furthermore, we illustrate the use of the technique in three different case studies investigating $NH_3$ emissions from agricultural sources. Additionally, we provide the



first description of plume height estimations obtained from the ground and column NH₃ concentration measurements. This study's results will also be valuable when using the SOF for other species and in other applications.


## 2. Instrument, flux quantification and measurement campaigns

### 2.1. SOF instrument and column retrieval

The SOF operation consists of recording solar infrared absorption spectra while driving through the gas plume (Fig. 1d and e). These spectra are captured by a solar tracker, containing several mirrors that transmit solar light to the spectrometer, following
the light as the car moves, and so there is a need for sunny or low cloud coverage conditions. Further, for spectra measurements, an FTIR instrument (Bruker IR cube ) is used, with a resolution of 0.5 cm⁻¹ and a dual detector InSb (Indium Antimonide, 2.5 – 5.5 µm) /MCT (Mercury cadmium telluride 9-14 µm). The detection limit for NH₃ columns with the SOF instrument calculated as $3\sigma$ is 2.2 mg m⁻² at a sampling rate of five seconds.

Alkanes are detected in the "C-H stretch band" at approximately 3.3 µm, while alkenes, propene and NH₃ are detected in the
"fingerprint region" at around 10 µm. The specificity of NH₃ is strong because this species' absorption at the fingerprint region is unique, with sharp absorption features well-separated from other species (Fig. 1c). The retrieval of NH₃ was initially conducted in a narrow spectral window (940 - 970 cm⁻¹) and subsequently in a broader window (900 – 1000 cm⁻¹). The broader window results in a more stable retrieval of the atmospheric background, although with slightly increased spectral noise. The calculated column values represent the relative abundance compared to a reference spectrum recorded outside the plume (Fig.
1a). Ideally, a location with low gas concentration should be chosen as the reference. In case of a noisy measurement, a posterior re-evaluation can be performed with a new reference spectrum. While retrieval of absolute columns is possible, it results in lower signal-to-noise ratio. The challenge with spectral retrieval is the long atmospheric path length of the solar spectra, which is affected by the strong absorptions of $H_2O$ and $CO_2$ in the atmosphere; therefore, other interfering species are taken into account. Retrieval is performed by fitting a calibration spectrum from the HITRAN (Rothman et al., 2005) infrared
database to simulate absorption spectra of the atmospheric background, using nonlinear multivariate analysis, and then calibrated according to pressure and temperature (Fig. 1b). The retrieval process is executed by a custom software (Kihlman, 2005). For reference, the fitting procedure is described in more detail in Mellqvist et al. (2010).

Ideally, each SOF-measured transect should ideally be recorded instantaneously, allowing the wind and turbulence conditions to be "frozen" in time. However, in practice, transects are carried out over a period ranging from a few seconds to minutes.
This duration is influenced by the distance to the source, the size of the plume and the road characteristics, factors which inherently introduce uncertainties to the measurement.





**Fig. 1: a) Example of spectra measured in the plume and in the background. b) Measured and fitted absorbance spectra and the calculated residual spectra. c) NH₃ calibration absorbance used to model the fitted spectra (approx. 40 mg m⁻³). d) Example of solar spectral measurements when crossing the target plume. e) Example of a box measurement around a target farm.**



## 2.2. Emission quantification

### 2.2.1. Emission Calculation

The gas flux, also generally interpreted as the emission from the source, is initially derived by integrating measured column
concentrations across the plume, following which the integrated mass of the target gas species can be obtained (Eq. 1). To
further calculate the flux, this integrated mass is multiplied by the wind speed parameter, $u_t$ (m s$^{-1}$) Eq. (1).

$$E_{NH3}(mg\,s^{-1}) = u_t(m\,s^{-1}) \int_P Column_{NH3_l}(mg\,m^{-2}) \cdot \cos(\theta_l) \cdot \sin(\alpha_l)\,dl(m) \qquad (1)$$

where p is the transect path across the plume, l corresponds to the travel distance and α is the angle between the wind and the
driving direction. The slant angle of the Sun is compensated for by multiplying the concentration with the cosine factor of the
solar zenith angle θ.

### 2.2.2. Determining the wind speed parameter

The wind is a vital part of SOF emission quantification (Eq. 1), and it should ideally correspond to the speed of the plume.
However, wind speed measurements are not straightforward, as the wind is disturbed close to the ground and changes according
to its height above the surface. Therefore, an approximation of the plume speed to be used as $u_t$ is the average integrated wind
profile (IWP$_{avg}$, Eq. 2) from ground to plume height (Fig. 2b). An assumption applied here is that the plume is vertically well
mixed, meaning a similar concentration from ground to plume height, which is usually the case during sunny conditions.
Additionally, in very unstable atmospheric conditions, the wind speed gradient is smoothed out by convection (Fig. 2a).
The IWP$_{avg}$ is obtained using Eq. 2, where H$_p$ is plume height (Section 2.2.3) and $u_z$ is horizontal wind speed (m s$^{-1}$) measured
at the different heights (z).

$$IWP_{avg} = \frac{\int_0^{H_p} u_z \cdot dz}{H_p} \qquad (2)$$




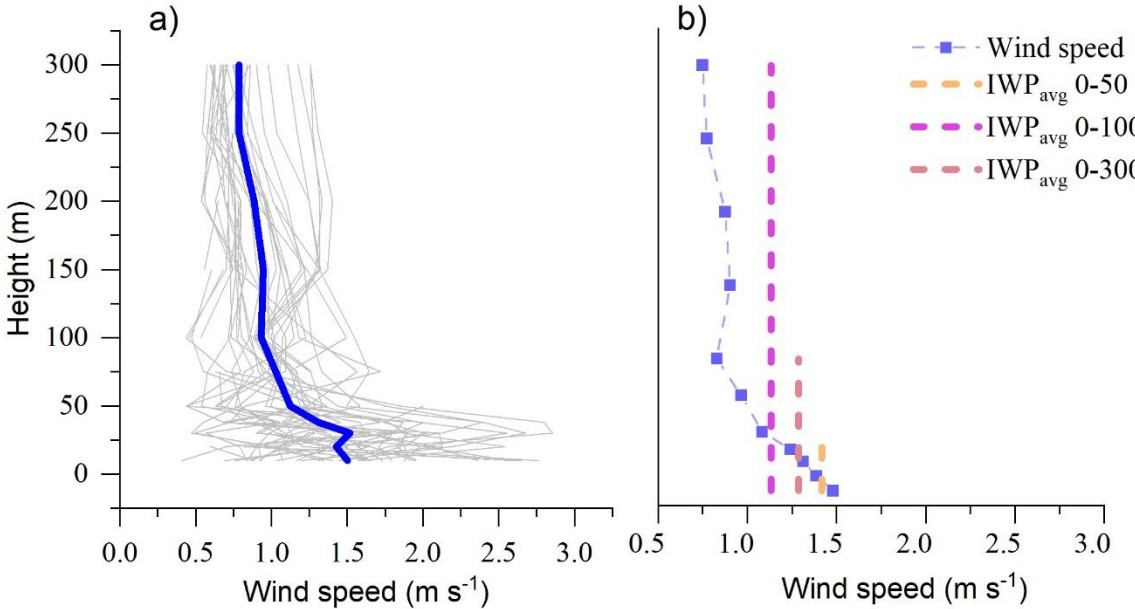


**Fig. 2: a) Wind profiles during case study C3, the grey lines show the individual profiles measured every 20 seconds and the blue line is the 10 minutes average. b) An example of integrated wind profile (IWP$_{avg}$) at three different height intervals (0-50, 0-100, 0-300 m) for a typical 5 minute average wind profile.**

**2.2.3. Plume height (H$_P$)**


To obtain the source's plume height, a novel approach is proposed, which involves using the ratio between the vertical column (mg m$^{-2}$) and the ground concentration (mg m$^{-3}$) of NH$_3$ (Eq. 3). The results of this estimation are demonstrated in case study 3 (C3). This method relies on the assumption that the plume is well mixed vertically (Fig. 3, Case I). However, in reality, the plume might not disperse homogenously (Fig. 3 Cases II or III), which brings uncertainty to the estimation, so it is considered an approximate assessment of H$_p$. For instance, when the plume is aloft (Fig. Case II), this methodology produces an unrealistically large plume height, in contrast to case II, which will be the opposite situation. Generally, solar insulation is strong during SOF measurements, which drives rapid vertical mixing and a plume dispersion like in Case I.


The NH$_3$ column (mg m$^{-2}$) was obtained by the SOF, while mobile extractive FTIR (MeFTIR) was used to measure ground NH$_3$ concentrations (mg m$^{-3}$). The latter instrument consists of an optical multi-path cell connected to a heated, temperature-controlled FTIR instrument (Galle et al., 2001; Vechi et al., 2023). In more detail, H$_P$ is calculated by integrating the ground concentration and column while crossing the plume path l, where θ is the solar zenith angle (Eq. 3). This method is referred to herein as the "vertical column ground concentration" (VCGC) ratio. Furthermore, the H$_p$ is calculated from the median of multiple transects.




$$H_p = \cos(\theta) \frac{\int Column_{NH3}(l)dl(\frac{mg}{m^2})}{\int Concentration_{NH3}(l)dl(\frac{mg}{m^3})} \qquad (3)$$

Alternatively, a rougher estimate of the $H_P$ might be obtained from a simpler calculation (Eq. 4), considering horizontal wind speed ($u_z$) at the available height, distance away from the emission source to the measurement road (P) and the speed at which the plume rises ($\sigma_w$) (m s$^{-1}$). Airborne measurements in Texas (Mellqvist et al., 2010) showed that the effective speed at which 150 the plume rises from industry in sunny conditions corresponded to 0.5 to 1 m s$^{-1}$, i.e. approximately the typical standard deviation of vertical wind (Tucker et al., 2009). Similar vertical wind data, i.e. ~0.5 m s$^{-1}$, were measured using a LIDAR instrument in C3, referred to herein as "plume transport vertical speed" (PTVS).

$$H_p = \frac{P\,(m)}{u(m\,s^{-1})}\sigma_w(m\,s^{-1}) \qquad (4)$$


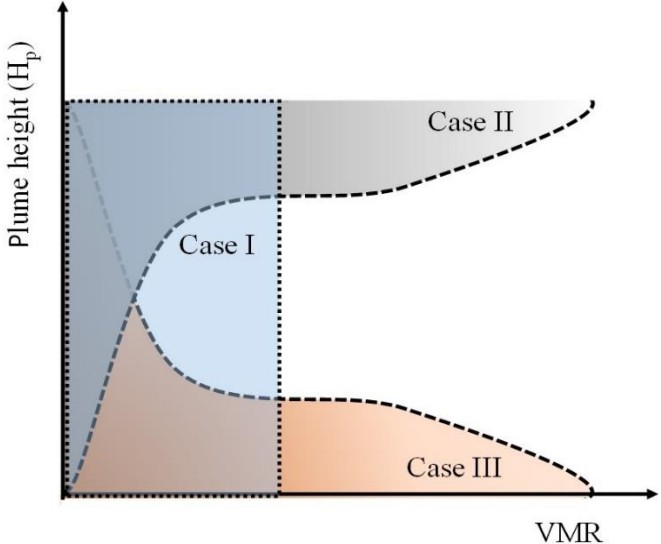

**Fig. 3: An illustration plume dispersion cases that can affect the plume height calculation. The y-axis represents the plume height, while the x-axis represents the volume mixing ratio (VMR). Case I: Ideal scenario. Case II: $H_p$ will be overestimated. Case III: $H_p$ will be underestimated.**


**2.3. Campaign description**

The SOF method was tested in a controlled release experiment and then demonstrated in three campaigns, each measuring NH$_3$ emissions from livestock production. The campaigns took place in France, the US (California) and Denmark, namely countries with extensive agriculture production and significant differences in manure management and climate conditions. The



campaigns were divided according to differences in wind measurements, the size of the target source and the interference of
nearby sources.

Wind measurements were generally recorded close to the source (~ 100 m), except for C3, when a wind LIDAR was positioned
approximately at five km away. Most campaigns used a 2D sonic anemometer (WXT510, Vaisala) or a vane wind monitor
(Model 05103, Young) mounted, respectively, on a ten- and/or three-metre mast. These 2-D wind sensors quantified horizontal
wind speed and direction.

*SOF validation – controlled release test (Grignon, France)*

A "blind" controlled release was performed over three days at a site in Grignon (France) to verify the accuracy of the SOF
method for $NH_3$ emission quantification (Supplementary information (SI), Fig. S1). Four release episodes were carried out by
Ineris at release rates varying from 0.48 to 1.1 kg h$^{-1}$. Gas was released from a pure $NH_3$ cylinder (Air Liquide - 84 litres -
purity > 99,99 %), equipped with a pressure regulator and a critical orifice (micrometric valve) to ensure a constant flow. The
$NH_3$ cylinder was set on a high-precision scale (Mettler Toledo, range 1-100 kg, precision 2 g) to control the stability of the
gas flow against time during the release. However, due to condensation and icing forming on the cylinder during release, the
final release flow values were assessed by weighing the cylinder prior to and after release, after the complete evaporation of
the condensed moisture. Horizontal wind speed and direction were measured at three and ten metres in height, using a vane
wind monitor and the 2D sonic anemometer, respectively. Information on meteorological conditions such as temperature,
relative humidity, precipitation and wind speed are shown in the Supplementary Information (SI Fig.S2). The transect SOF
measurements were conducted by Fluxsense downwind of the release at average distances of 150 - 300 m. Release rates were
unknown to the SOF operators until the final results were submitted, in order to ensure a proper "blind test" validation.


*Case study 1 (C1) – Pig and dairy farm (Denmark)*

Case study 1 consisted of a two-day measurement campaign at two small-scale animal farms in Denmark, each of which was
well-isolated from other interfering sources. $NH_3$ emissions were measured at a pig farm (C1a) and a cattle farm (C1b), and
transects were performed at 250 and 900 m, respectively. The pig farm housed approximately 600 sows with piglets and
weaners, while the cattle farm had approximately 700 dairy cows, plus heifers and calves. Horizontal wind speed and direction
were obtained from two vane wind monitors placed on three- and ten-metre-high masts. Columns were measured downwind
from the farms, while upwind fluxes were measured only once or twice because there were no other interfering sources.

*Case study 2 (C2) - Dairy complex (USA, California)*

In case study 2, the SOF method was used to measure $NH_3$ emissions on a large dairy complex in Chino (California), a sizeable
and concentrated area (21 km$^2$) without other important $NH_3$ sources. Transects were collected in one day and were performed
around the farm's fence line area, comprising a distance of 18 km for one transect. The area housed approximately 36,000
heads (CARB, personal communication 2015). One vane wind monitor performed wind measurements on a 10 m mast, and



these were done by encircling the area; therefore, emissions were calculated by estimating the flux leaving the area minus the
one entering it.

*Case study 3 (C3) – Dairy concentrate animal feeding operations (USA, California)*

Lastly, case study 3 was conducted in dairy concentrated animal feeding operations (CAFOs) in the San Joaquin Valley (SJV),
California. The results present the combination of the SOF (column) and the MeFTIR (ground concentration) instruments to
demonstrate plume height calculations using the results from this case study. These were sources with large emissions, placed
in high farm-density areas. $NH_3$ measurements were done at the farms' fence line, approximately 1 km from the source, for
one or two days for SM1 (C3a) and SM2 (C3b), respectively. A wind LIDAR was used here, its detection principle based on
the Doppler shift of an infrared pulse (~1.5µm) emitted by the instrument, which is then reflected by atmospheric aerosols.
The instrument used in this campaign (Campbell Scientific, LIDAR ZX300) provided horizontal and vertical wind speeds and
directions ranging from 10 m to 300 m above ground at 11 different heights. In this case study, the $IWP_{avg}$ was used as a wind
parameter ($u_t$) for the emission calculations, averaged at five-minute and three height intervals, i.e. 0-50 m, 0-100 m and 0-300
m. Upwind and downwind measurements were necessary to isolate emissions from the individual farm, due to other interfering
sources near the target farms.

### 3.   SOF uncertainty methodology

This study establishes a methodology for quantifying the uncertainty associated with Solar Occultation Flux (SOF)
measurements based on the Guide to the Expression of Uncertainty in Measurement (GUM) method (Joint Committee For
Guides In Metrology, 2008). This marks the first time that uncertainties in $NH_3$ SOF emission measurements from livestock
production have been established, albeit drawing from principles outlined in the European measurement standard for VOC
monitoring of refineries (CEN EN 17628 European standard, 2022). The investigation identifies and sums both random and
systematic uncertainties to establish a total standard 68% confidence interval (CI 68%) or expanded uncertainty (CI 95 %). It
should be noted that most scientific articles, including past SOF studies (Johansson et al., 2013; Kille et al., 2017; Mellqvist
et al., 2010), only consider standard uncertainties (CI 68 %). This paper, however, adopts a more comprehensive approach in
line with industry and metrology institutes (Joint Committee For Guides In Metrology, 2008). As part of the uncertainty,
description this study proposes a novel method to assess spectroscopic uncertainties, demonstrating superior results compared
to the approach typically used in general spectroscopic measurements.

Measurement random uncertainty is caused by many factors, with wind turbulence the most significant contributor. This
uncertainty decreases in line with the number of samples taken; hence, the SOF European standard for refinery measurements
recommends a minimum of 12-16  transects divided over several days (CEN EN 17628 European standard, 2022) for this type
of source. In turn, systematic errors will persist, independently of the number of transect. They are often correlated to the
technique, instrumentation and measurement of other important variables, such as wind speed, and in this case, establishing



best practices is one way to reduce them. The measurement uncertainty methodology is combined with data quality requirements, which must be fulfilled for valid measurements. This includes sufficient solar height, relatively persistent wind direction and speed above 1.5 m s$^{-1}$ and sufficient measurement quality.

### 3.1. Spectroscopy uncertainty

Systematic spectroscopy errors can be divided into two categories, namely errors due to uncertainty in the strength of the absorption cross-section and imperfect spectroscopic fitting of the band shapes. Absorption strength uncertainty ($U_{abs\text{-}NH3}$) of 2 % ( $|(I_{obs} - I_{cal})/ I_{obs}|$ ) for the $NH_3$ cross-section was found by Kleiner et al. (2003) for the full band of 700 to 1200 cm$^{-1}$. Therefore, it ($U_{cros}$) was calculated using absorption strength ($U_{abs\text{-}NH3}$) (Kleiner et al., 2003), further divided by 1.96, as this error was considered a normal distribution (Eq. 5).


$$U_{cros} = \frac{U_{abs-NH3}}{1.96} \tag{5}$$

Imperfect spectroscopic fitting can have different causes, for instance errors due to the shape of the reference cross-sections used, wavelength shifts or errors in instrument line shape characterisation. Consequently, the spectroscopic fitting routine 245 cannot account perfectly for all spectroscopic absorption features and may systematically over- or underestimate column. The fitting residual, defined as the difference between measured and fitted absorbance, captures some information regarding the total fitting error. The root-mean-square of the residual (RMS) is a commonly used measure of the fitting error magnitude, which can be used to estimate column uncertainty caused by fitting errors. Therefore, to assess the retrieval error ($U_{ret}$), we calculated the ratio between average $NH_3$ absorbance in 960 to 968 cm$^{-1}$ (AVG-abs$_{960\text{-}968\mu m}$) (Fig. 1b) and the standard deviation 250 of the fitting residual (STD) in the same wavelength range, divided by the square root of the number of points (Eq. 6). The ratio was calculated for measurement points inside and outside the plume, and the linear regression curve's slope was considered as the error.

$$U_{ret,1} = \left( \frac{\frac{STD}{\sqrt{n}}}{\overline{abs}_{(960-968)\mu m}} \right) \tag{6}$$


Previous studies (Griffith, 1996) have estimated the fitting uncertainty as

$$U_{ret,2} = \frac{STD}{\overline{abs}} \tag{7}$$

Additionally, we estimated uncertainty based on dividing the integrated area under the fitting residual $A_r$ with the integrated area under the fitted $NH_3$ absorption $A_{abs}$.



$$U_{ret,3} = \frac{A_r}{A_{abs}} \tag{8}$$

In this study, different estimates were investigated by deliberately introducing errors into the fitted cross-sections and using

these cross-sections in a spectral fit applied to a synthetic spectrum with absorption from a known column. Different uncertainty estimates (Eqs. 6, 7 and 8, SI Fig. S3, S4 and S5) were then calculated based on the residual from the fitting and compared to the error in the fitted column. The cross-sections included three error types: resolution error, shifting error and a multiplicative Gaussian noise error. For each case, a random error was chosen from each of the three types of errors within a specific range. The resolution error was a scaling factor in the range of one to four, the wavelength shift error was an offset in

the range -0.2 to 0.2 cm$^{-1}$ and multiplicative Gaussian noise had a standard deviation from 0 to 0.1. In total, 1000 random such as these were conducted, and Fig.4a shows the resulting uncertainty estimates and column errors for each case. Fig. 4b provides an example of the fitted NH$_3$ absorbance and residual for one of these cases. The uncertainty estimate in Equation 7 was found to significantly overestimate the column error. In contrast, the uncertainty estimate in Equation 6 was a better estimate, with the error being smaller than this estimate in roughly 95 % of cases. The uncertainty estimate based on the area (Eq. 8) was

determined to significantly underestimate column errors in most cases.

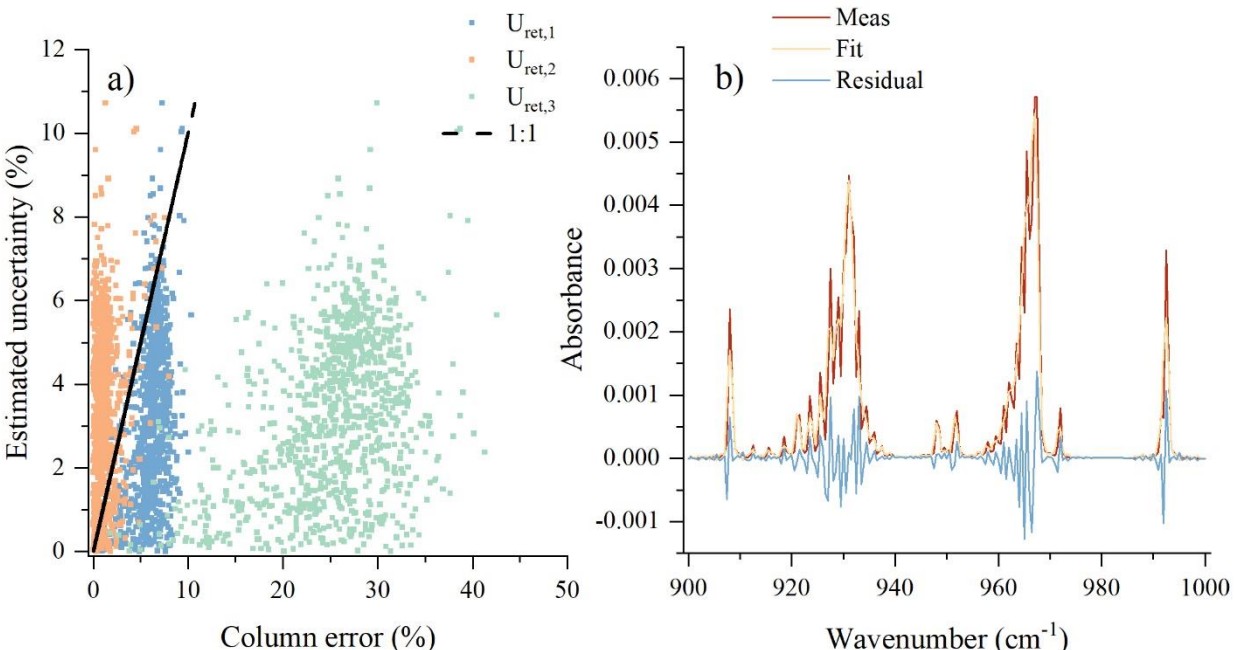

Fig. 4: a) Column errors and uncertainty estimates for 1000 simulated test cases. Uncertainty estimate from Eq. 6 in green, from Eq. 7 in blue and from Eq. 8 in orange. b) Example of the fitted NH$_3$ absorbance for one of the simulated cases.




### 3.2. Background uncertainty

Background might differ on a systematic way on either side of the emission plume. Among other things, this might indicate the presence of a secondary source on the side or upwind of the target source (Fig. 5) or the influence of interfering background species when the solar angle changes. Background uncertainty ($\pm U_B$) corresponds to the relative difference in flux when

choosing either the left or the right value as the assumed background. As the background value changes within the plume and is unknown, the uncertainty distribution is considered to be rectangular. Therefore, to obtain the standard uncertainty, it should be divided by the square-root of three according to GUM (Joint Committee For Guides In Metrology, 2008) (Eq. 9).

$$U_B = \frac{(\Delta col_{Background})/2}{\sqrt{3} \cdot A_{col}}$$ (9)


Here $\Delta_{col}$ corresponds to the difference in the measured columns on either side of the allocated emission plume while $A_{col}$ corresponds to the integrated column area across the plume.

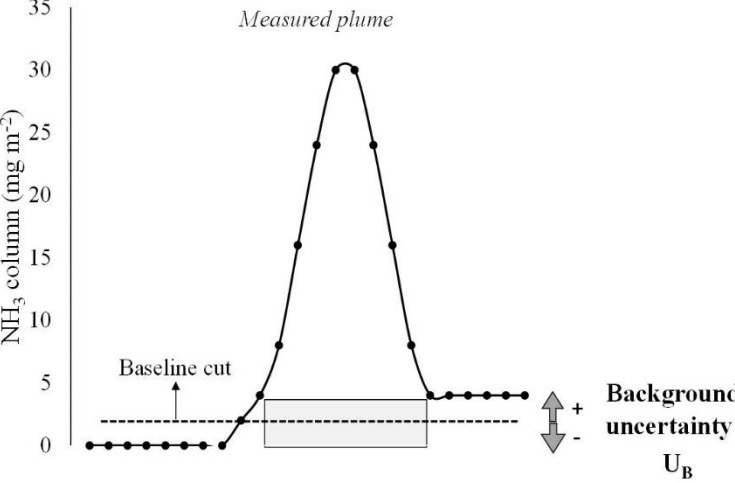

**Fig. 5: Assessment of systematic background uncertainty. The grey-shadowed box represents the uncertainty area being added to the quantification.**

### 3.3. Wind speed uncertainty

Wind speed is the largest source of uncertainty in SOF measurements (Johansson et al., 2013; Kille et al., 2017; Mellqvist et

al., 2010). The wind speed parameter ($u_t$ in Eq. 1) should be an approximation of the plume speed, in which case the IWP$_{avg}$ is the best estimate of this parameter. Sunny, convective conditions smooth out wind gradient convection, which together with



$H_p$ estimation helps minimise errors. For the different case studies, the plume height was estimated according to the available information (Table 1), and only in case study C3 was the plume height measured (VCGC, Eq. 3).

In the validation test and C2, two wind masts held the wind monitors, one at 3 m and the other at 10 m. The wind profile was obtained by estimating the α factor using Eq. 10, where U is a known wind speed at two different heights. Thereafter, the obtained α factor was used to estimate the wind speed at the plume height (Eq. 11). Further, the $IWP_{avg}$ was obtained using Eq. 2, by using the estimated wind profile. Furthermore, uncertainty was estimated by the difference between the measured wind speed (10 m), which was used for the flux calculations and the estimated $IWP_{avg}$ from ground to $H_p$ (Table 1, Eq. 12). For C1, only one 10 m mast was used to measure the wind, so we estimated the error of choosing different vertical profiles by using information from another study at the same geographic location. Moreover, in C3, we had a LIDAR as a wind sensor, so the $IWP_{avg}$ was directly calculated at different height ranges (Eq. 2). Since the wind speed profile was actually measured instead of estimated, the error estimation in C3 is a better prediction of wind speed error (Table 1, Eq. 12).

$$\alpha = \frac{\log(U_2/U_1)}{\log(z_2/z_1)} \tag{10}$$

$$U_{(z)} = U_2 \left(\frac{z}{z_2}\right)^{\alpha} \tag{11}$$

**Table 1: Parameters used in calculating wind speed error uncertainty.**

|  | **Validation** | **Case study C1** | **Case study C2** | **Case study C3** |
|---|---|---|---|---|
| **Wind speed data ($u_t$)** | 10 m | 10 m | 10 m | Measured $IWP_{avg}$ (0-50, 0-100, 0-300 m) |
| **Plume height ($H_P$)** | Estimated (Eq. 4) | Estimated (Eq. 4) | Estimated (Eq. 4) | Measured (Eq. 3) |
| **Integrated Wind profile ($IWP_{avg}$)** | Estimated (Eq. 2, 10 and 11) | Estimated (Eq. 2, 10 and 11) | Estimated (Eq. 2) using C3 data | Measured (Eq. 2) |
| **Error estimation (Eq. 12)** | $U_{wind} = \dfrac{\left(1 - \dfrac{IWP_{Avg}}{u_t}\right)}{1.96}$ | $U_{wind} = \dfrac{\left(1 - \dfrac{IWP_{Avg}}{u_t}\right)}{1.96}$ | $U_{wind} = \dfrac{\left(1 - \dfrac{IWP_{Avg}}{u_t}\right)}{1.96}$ | $U_{wind} = \dfrac{\left(\dfrac{IWP_{Avg}\ (0-300)}{IWP_{Avg}\ (0-50)}\right)}{1.96}$ |

In this study, the uncertainty associated with wind direction was not factored into our measurements due to our knowledge of the source's precise location. This understanding allowed us to make necessary corrections to the wind direction, assuming that the emission plume moves uniformly from the known source. These corrections were based on visual observations made by the data processing operator. However, when the source location is not accurately known, it is crucial to consider and incorporate the uncertainty related to wind direction into the analysis. This approach aligns with the procedures followed in other SOF assessments when dealing with similar uncertainties (Johansson et al., 2014).



### 3.4. Calculation of standard and expanded total uncertainty

In each case study, random uncertainties, $U_{rand}$, were calculated as the standard error of the mean of the measured gas flux, as demonstrated by Eq.13. The total variability is affected by the random variabilities of all the individual parameters that are used in the flux calculation according to Eq.1. The overall random uncertainties decreases in an inverse proportion to the square root of n.

$$U_{rand} = \frac{(STD)}{\sqrt{n}} \tag{13}$$

For each case study, the systematic and random uncertainties were combined in a root-sum-square, resulting in the standard uncertainty (CI 68 %). Furthermore, effective degrees of freedom were considered, and expanded uncertainty (CI 95 %) was also calculated. Calculations followed the GUM methodology (Joint Committee For Guides In Metrology, 2008) using Eq. (14), where $U_{tot}$ is total relative uncertainty and k is the coverage factor (ranging 1.96 – 3.00), depending on the degrees of freedom, N, and the confidence interval.

$$U_{tot} = k\sqrt{\left(U_{cros}{}^2 + U_{ret,1}{}^2 + U_B{}^2 + U_{wind}{}^2 + U_{rand}{}^2\right)} \tag{14}$$

## 4. Results

### 4.1. Uncertainty analysis

Each estimated uncertainty for the different case studies, as well as for the validation study, is shown in Table 2. Expanded uncertainty (CI 95 %) ranged from 15.1 to 37.4 %, with a median value of 27 % for all case studies.

Here the systematic wind uncertainty, $U_{wind}$, represents one of the largest sources of errors (Table 2) while wind turbulence contributes significantly to the random uncertainty. The estimated $U_{wind}$ was particularly high in C1b and C2 because of the relatively high $H_P$ (130 - 500 m), which was estimated by the PTVS method (Eq. 4), while wind information was obtained at 10 m high, thereby limiting the available field instrumentation. In contrast, in C3, despite the large $H_P$ (400 m), wind speed measurements were done using a LIDAR, which gathers data up to a height of 300 m, resulting in an $U_{wind}$ smaller than at C1b and C2. Additionally, in C3, the $H_P$ could be better estimated than in the other campaigns using the VCGC method (Eq. 3), which resulted in a decrease in $U_{wind}$ and, consequently, total uncertainty. $H_P$ is discussed in more detail in the following section. Moreover, for most case studies, a large number of transects was recorded, therefore, the random uncertainty, $U_{rand}$, was low. The exception was C2, which only had three transects, although these resulted in similar fluxes and therefore a low random uncertainty.





**Table 2: Overview of estimated uncertainties and validation and in the other case studies.**

|  | Validation | C1a | C1b | C2 | C3a | C3b |
|---|---|---|---|---|---|---|
| *Systematic* – $U_{cros}$ (%) | 2.0 | 2.0 | 2.0 | 2.0 | 2.0 | 2.0 |
| *Systematic* – $U_{ret,1}$ (%) | 4.4 | 4.4 | 4.4 | 4.4 | 4.4 | 4.4 |
| *Systematic* – $U_b$ (%) | 1.8 | 5.0 | 9.0 | 0.9 | 1.5 | 0.4 |
| *Systematic* – $U_{wind}$ (%) | 3.0 – 6.0 | 3.0 | 32.0 | 23.5 | 11.0 | 11.0 |
| *Systematic* – Gas release (%) | 2.0 | NA | NA | NA | NA |  |
| *Random* – $U_{rand}$ (%) | 3.3 – 6.9 | 9.0 | 7.1 | 4.6 | 9 | 12 |
| **Standard uncertainty (CI 68 %)** | **6.5 – 8.7** | **10.6** | **19.1** | **13.6** | **12** | **14** |
| **Expanded uncertainty (CI 95 %)** | **12.7 – 17.5** | **21.0** | **37.4** | **27.0** | **25** | **29** |
| Estimated $H_p$ (m) | 11 – 40 | ~30 | ~130 | > 500 | ~ 500 | ~ 400 |

### 4.1.1. Plume height ($H_P$)

The MeFTIR and SOF were operated simultaneously in the vehicle, making it possible to estimate the plume height ($H_p$) using the VCGC method according to Eq. 3 and compare, this to the $H_p$ estimated using the PTVS method (Eq. 4). Figure 6a presents examples of $NH_3$ columns (left-axis) and ground concentrations (right-axis) measured in three distinct plumes (P1, P2, P3). In the first peak, P1, the ground concentrations were comparable to P2 (right-axis), while the column measurements were lower than P2 (left-axis), indicating that P1 was located close to the ground. Conversely, P2 was at a higher height. Similarly, for P3,

the columns (left-axis) were lower than P2. However, the ground concentrations (right-axis) were much higher, again suggesting a plume close to the ground (Fig. 6a). Furthermore, the second method (PTVS, Eq. 4) was utilized and compared to the VCGC method, showing slightly lower, but similar results (Fig. 6b). The VCGC method is more accurate as it does not require assumptions about the vertical plume speed. Additionally, in more complex cases, such as when the $NH_3$ source is spread and heterogeneous (as seen in farm 8), the PTVS approach did not yield values similar to those in the VCGC method

(Fig. 6b).



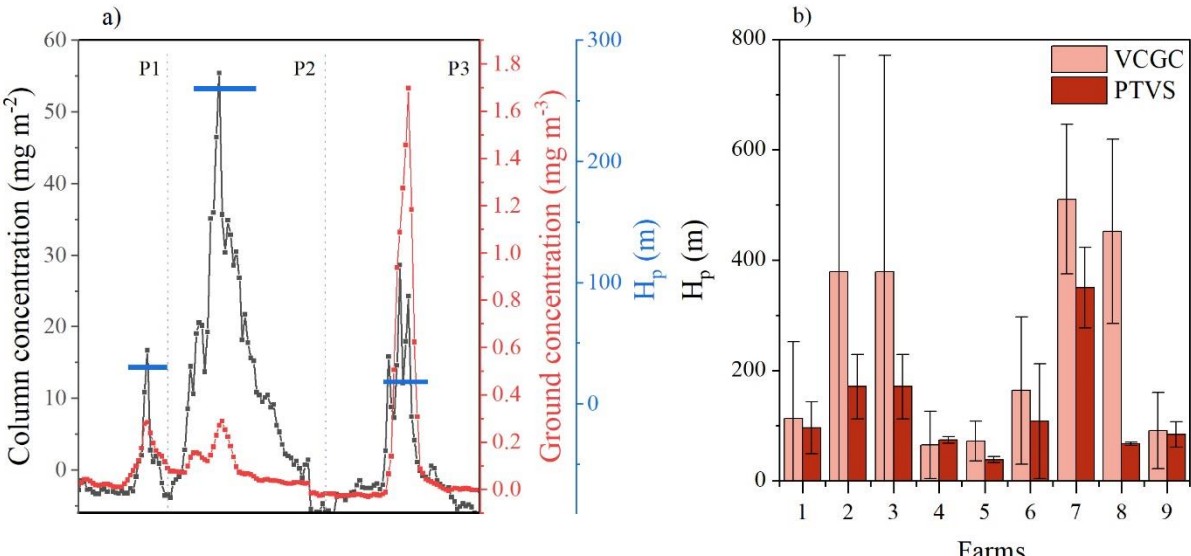

**Fig. 6: a) Simultaneous measurements of NH₃ columns and ground concentrations. P1 and P3 were ground sources. b) Examples of plume height calculation using the two methods (VCGC), (light red bar), the error bars correspond to the variation in the plume height calculation and the estimated values using vertical wind speed (PTVS) (dark red bar), the error bars correspond to the variation of the $H_P$ calculation (variations in wind speed and measured distance).**

### 4.2. Validation

In the NH₃ validation test, controlled gas releases varied from 0.48 to 1.1 kg h⁻¹, while SOF NH₃ quantified emissions varied from 0.41 to 1.27 kg h⁻¹ (Fig. 7, Table 3). On average, wind speed varied from 3.8 to 5.9 m s⁻¹, and the direction changed from weak north-easterly winds on 22-September to stronger and south-westerly winds on the two last measurement days. The weather conditions were sunny with low cloud coverage on 28-September and 1-October, while on 22-September the presence of clouds was more considerable, although measurements were still possible.



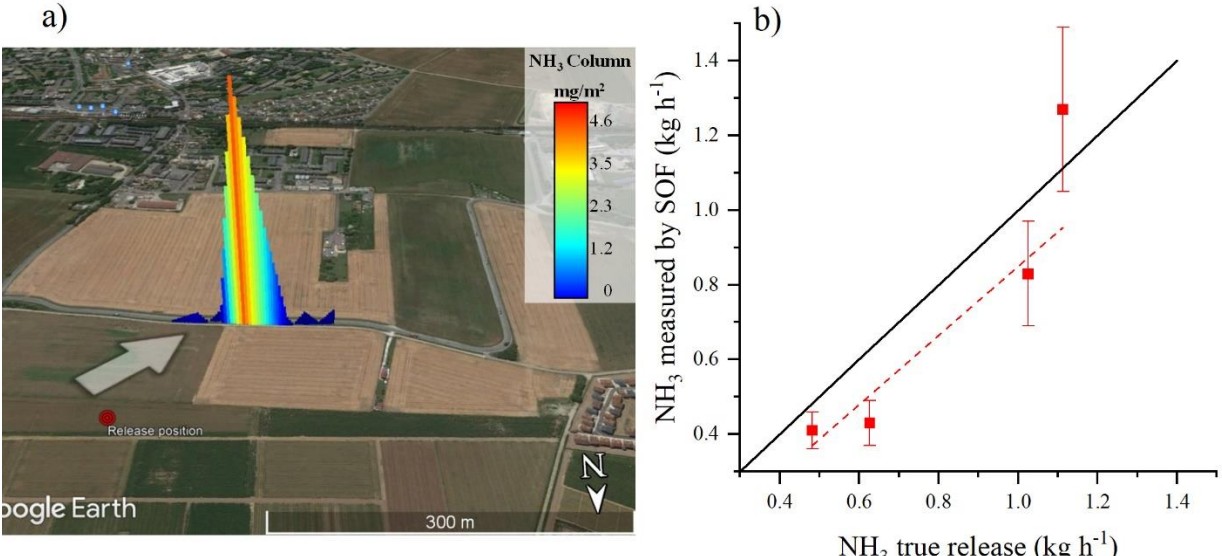

**Fig. 7: a) Example of measured plume on the day 22-September at 14:55. The red dot indicates the NH₃ release point, and the arrow**
**shows wind direction. b) Controlled release rates and SOF quantified rates (average ± expanded uncertainty (CI 95 %)). Map source:**
**© Google Earth**

The relative error was between a minimum of -31 % and a maximum of +14 % (Table 3). Additionally, the calculated standard

uncertainty (CI 68 %) ranged from 6.4 to 8.7 %, and the CI 95 % ranged from 12.7 to 17.5 % (Table 2). The estimated

uncertainty explained the error observed only in the first release (Table 3, Fig. 7b), albeit within a 5 % difference in the last

two releases (1-October). Potential sources of error include wind speed measurements, particularly as the estimated plume

height ranged from 11 to 40 m (Table 3), while wind data was collected at 10 m height. Although wind uncertainty was

considered in the budget estimation, the lack of vertical wind profile measurements may have introduced limitations to the

analysis.


**Table 3: Overview of the NH₃ validation experiment.**

| Date | Measurement distance (m) | Wind speed (m s⁻¹) – Direction | Number of transects | Controlled release rate (kg h⁻¹) | SOF emission (kg h⁻¹) | Error (%)ᵃ | Total expanded uncertainty (%) | Estimated $H_P$ (m) |
|---|---|---|---|---|---|---|---|---|
| 22-September | 180-320 | 3.8 - NE | 17 | 1.11 | 1.27 | 14 | 17.5 | ~ 40 |
| 28-September | 180-220 | 4.2 - SW | 34 | 0.63 | 0.43 | -31 | 12.7 | ~ 20 |
| 1-October | 150-180 | 5.8 - SW | 26 | 0.48 | 0.41 | -15 | 12.9 | ~ 12 |
| 1-October | 150-180 | 5.9 - SW | 22 | 1.03 | 0.83 | -19 | 17.2 | ~ 11 |

ᵃ Error estimated from: 100·(SOF emissions - Controlled release)/Controlled release.





In 75 % of the measurements, the NH$_3$ SOF quantifications were lower than the actual release, possibly due to NH$_3$ dry
deposition or gas temporary loss in the release system, such as trapping in ice. NH$_3$ dry deposition depends on factors such as
wind speed, source height, atmospheric stability, surface roughness length and surface concentrations (Asman, 1998).
However, a deep analysis of NH$_3$ dry deposition is beyond the scope. The measurements on 22-September exceeded the actual
release, potentially impacted by less than ideal cloud conditions during the campaign, affecting the light intensity measured.

**4.3. Case studies**

These case studies were utilized to validate the Solar Occultation Flux (SOF) method's effectiveness for measuring NH$_3$
emissions from various livestock production systems, in addition to assessing the real-world applicability of the developed
uncertainty methodology. A measurement overview is provided in Table 4, and specific transect examples from each
measurement campaign are depicted in Fig. 8. However, these emission data represent snapshots, confined to one or two days
of measurement, and thus do not offer a reflection of annual emissions

**Table 4: Overview of results for the SOF NH$_3$ measurements.**

| | C1a | C1b | C2 | C3a | C3b |
|---|---|---|---|---|---|
| Month | October | October | October | May | May |
| Distance from the center of source (m) | 220 | 800 | 2500 | 2000 | 1000 |
| Measurement interval | 09:40-14:30 | 12:10-16:20 | 13:30-16:00 | 12:20-14:00 | 14:20-17:30 |
| Number of transects | 20 | 14 | 3 | 7 | 13 |
| Avg. wind speed (m s$^{-1}$) | 3.1 | 3.1 | 4.0 | 3.0 | 5.7 |
| Number of animals | 600 sows | 700 cows | 36000[b] cows | -[a] | -[a] |
| Avg. emission (kg h$^{-1}$) | 1.1 | 2.2 | 245.0 | 166.0 | 142.2 |
| Uncertainty (CI 95 %) | **21.0** | **37.4** | **27.0** | **25.0** | **29.0** |
| Emission factor (g LU$^{-1}$ h$^{-1}$) | 2.4 | 2.5 | 6.8 | | |

[a] Unknown numbers. [b] Number of animals obtained from personal correspondence with the California Air and Resources
Board (CARB).






**Fig. 8: a) (C2) NH₃ columns measured at Chino, made by encircling the feedlots area in a box, the arrow indicates the wind. b) (C1a) Pig farm example (Total farm), flux on the figure corresponded to 0.55 kg/h. c) (C1b) Dairy farm plume example, corresponded flux of 2.52 kg/h. d) (C3) Example of measurement from individual CAFOs, on the upwind from the farm there was emissions from the field. Map source: © Google Earth**

### 4.3.1.    C1 - Small and isolated sources - Pig and dairy single farms (Denmark)

Emissions from small and isolated farms are challenging to measure, primarily because of their low emissions, and thus low concentrations, which are difficult to measure at a distance away from the farm. Total farm $NH_3$ emissions averaged $1.07 \pm 0.23$ kg h$^{-1}$ (CI 95 %) for pig farms (C1a, Fig. 8b). Thus, the SOF could measure concentrations as low as 1 kg/h with an uncertainty of ~ 21 %. Emissions were normalised by livestock unit (1 LU = 500 kg of body weight) to obtain an emission factor (EF) of $2.4 \pm 0.5$ g LU$^{-1}$ h$^{-1}$, while the literature has reported EFs of 1.88 g LU$^{-1}$ h$^{-1}$ for the house only (Rzeźnik and Mielcarek, 2016).





The dairy farm (C1b, Fig. 8c) had average emissions of $2.3 \pm 0.9$ kg h$^{-1}$, corresponding to an EF of $2.5 \pm 0.9$ g LU$^{-1}$ h$^{-1}$. Based

on the literature, EF dairy farm houses are expected to have around 1.1 g LU$^{-1}$ h$^{-1}$ for the house only (Rzeźnik and Mielcarek, 2016). However, uncertainty in relation to wind speed measurements was relatively high (U$_{wind}$ 32 %) due to limited wind instrumentation. Additionally, there is also the possibility of dry deposition, due to the large distance between the source and the road used for the measuring equipment (800 m). Moreover, the emission rates obtained for C1a and C1b offer only a brief snapshot of daytime emissions, making comparisons with existing literature somewhat uncertain. This issue points to a larger

uncertainty stemming from the inherent limitation of not capturing the full diurnal cycle. This factor can significantly impede effective comparison between different studies unless the data is normalized to a model predicting expected emissions across a full day-night cycle. It is important to note that this aspect relates more to the representativeness of the measurements, rather than any inherent issues with the measurement process itself.

### 4.3.2.    C2 – Box measurements of several sources – Dairy complex (USA)


In case study C2 (Fig. 8a), the SOF method was utilized to quantified NH$_3$ emissions from the Chino dairy complex in California, USA. Although the emissions magnitude was significant, the extensive size of the complex (18 km perimeter), necessitated almost an hour to measure one box transect. This prolonged measurement time, coupled with potential changes in wind speed and direction, could have contributed to an increased uncertainty in the measurements.

NH$_3$ emissions averaged $245.0 \pm 66$ kg h$^{-1}$, while the EF was 6.8 g head$^{-1}$ h$^{-1}$. In comparison with the NH$_3$ flux estimations for this area using retrievals from a satellite, Infrared Atmospheric Sounding Interferometer (IASI) (Van Damme et al., 2018), emissions were similar to SOF, 4.3 g head$^{-1}$ h$^{-1}$, ranging from 1.1 - 51 g head$^{-1}$ h$^{-1}$. In contrast, other studies showed larger EFs as 18.5 to 42 g head$^{-1}$ h$^{-1}$ (Leifer et al., 2017, 2018) and 14.9 to 79.7 g head$^{-1}$ h$^{-1}$ (Nowak et al., 2012). High fluctuations in NH$_3$ emissions are expected because they depend on meteorological factors (wind speed, temperature, solar radiation),

although some variability might also result from the different techniques used. Here, the estimated measurement uncertainty was 27 %, with the U$_{wind}$ being the largest source of errors.

### 4.3.3.    C3 – Large source surrounded by other sources – Dairy CAFOs (USA)

One of the challenging types of facilities for the solar occultation flux (SOF) to measure are large-scale, individual farms in

areas of high farm density. The primary difficulty stems from interference from surrounding sources near the target farms. As such, upwind or box measurements, which encircle the source, were required to isolate the farm being measured (Fig. 8d).

Dairy CAFOs averaged 142 kg h$^{-1}$ for C3b and 165 kg h$^{-1}$ for C3a. The number of animals was not known; hence, emission factors (EFs) could not be established. Nevertheless, ammonia (NH$_3$) emission rates and EFs from these kinds of facilities have been documented elsewhere, (Vechi et al., 2023).



In C3, the $IWP_{avg}$ and $H_p$ were measured differently from the other campaigns, i.e. estimated based on more uncertain calculations. Total expanded uncertainty ranged from 25 % to 29 %, and although $U_{wind}$ was lower than the other campaign (11 %), random uncertainty made a large contribution (9 % to 12 %).

## 5.   Conclusions and method application perspective

$NH_3$ emissions are challenging to quantify due to their high stickiness, which makes it difficult to sample without losses.
Additionally, in the case of diffuse emissions from farms, $NH_3$ quantification might be hampered by interference from fertiliser application and transport emissions or by dry deposition, meaning that concentrations are lost within a few metres from the source. These issues must be considered when designing and applying new instruments and methods. The SOF method has advantages for $NH_3$ quantification because it offers a contact-free measurement, thereby avoiding issues with gas adsorption into the gas inlet and instrument interior. Additionally, it has a fast time response (~5 s) which, when combined with the
flexibility provided by the mobile platform, helps cover large areas over a measurement day. Furthermore, the SOF measures vertical columns, which is advantageous compared to ground concentrations, as the latter might be affected by $NH_3$ deposition (Lassman et al., 2020). Moreover, SOF column measurements can be used to validate satellite column measurements, as has been recently done (Guo et al., 2021). Estimating measurement uncertainty is essential because it indicates measurement precision; therefore, when comparing the obtained rates with other literature and models, uncertainty can better indicate
whether or not values are significantly different.

Nonetheless, measurements using the SOF method are limited by required weather conditions, such as sunny skies and low cloud cover. As such, nighttime and heavily cloudy weather are not suitable for measurements. Additionally, the solar angle required for measurements leads to limitations for winter measurements at certain latitudes. $NH_3$ emissions are higher during daytime and sunny conditions, so, when using this method to estimate annual emissions or to compare to other studies and
inventories, any diurnal emission variation must be considered (Lonsdale et al., 2017; Zhu et al., 2015a). This can be done by using models that estimate daily $NH_3$ variations, using meteorological information or other parallel measurements.

Here, the validation test and case studies have shown the SOF method's applicability and the accuracy level that the method can reach when best practices are followed. This study demonstrates that the wind speed vertical profile is a crucial parameter, which is more easily measured using LIDAR instrument. Additionally, to improve measurement accuracy and the choice of
wind parameters, plume height should be estimated by combining measurements of ground and column. Furthermore, the technique was demonstrated to be suitable for large, concentrated areas and smaller sources with emissions as low as 1 kg/h, obtaining an uncertainty level ranging from 21 to 37 %, with a median value of 27 %. This study shows the potential of SOF technique for a better quantification of diffuse $NH_3$ emissions related to livestock buildings, a source which is still poorly known.


*Data availability*



Data can be provided by the corresponding authors upon request.

*Author contributions*

JM, NV, MD, FG, JJ, BO, JS, SB planned and executed the measurement campaign; JM, NV and JJ developed the uncertainty methodology; JS, BO, NV, JJ analyzed the data; JM and NV wrote the manuscript draft; JM, NV, CS, MD, FG, BO and JJ reviewed and edited the manuscript.

*Competing interests*

The authors declare that they have no conflict of interest.

*Acknowledgements*

We would like to thank California Air and Resources Board for the sponsorship in collecting part of the data (Case Study 3)
in the contract 17RD021 "Characterization of Air Toxics and GHG Emission Sources and their Impacts on Community-Scale Air Quality Levels in Disadvantaged Communities" by FluxSense Inc. Additionally, we thank to AgroParisTech in Grignon for supporting the study by providing a place for the Validation campaign.

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
