# Peer review of "An uncertainty methodology for solar occultation flux measurements: ammonia emissions from livestock production"

_EGUsphere, 2023_

## Referee Comment (RC1)

**An uncertainty methodology for solar occultation flux measurements: ammonia emissions from agriculture**

By Johan Mellqvist et al.

The authors give a detailed description of the issue of measuring ammonia (NH3) emissions from the agricultural sector. Using the solar occultation flux (SOF) method, which is a remote sensing technique with mobile FTIR, the conducted measurements of NH3 from a validation experiment and three case studies are used in combination with different combinations of wind measurements and an in situ FTIR to infer NH3 emission fluxes and their uncertainties.

The paper is appropriate for publication in AMT because it provides a new methodology to infer plume height prior to estimating emission fluxes. However, the specific comments in the next section need to be addressed by the authors beforehand.

**Specific Comments:**

1. The structure of the paper is not well outlined:
   a. In section 2 the authors introduce a validation and three case studies, but then in the results section are not very specific about the order of how these were introduced. This leads to confusion when e.g. in section 4 it becomes less clear that MeFTIR was only used during C3. It is also confusing when in the results section suddenly specific plume transects are discussed but it is not clear which case study these relate to. I suggest restructuring section 2 and better introducing the measurements and studies. It would help to have a designated subsection for the wind related instruments as well as one for the MeFTIR such that it doesn't first appear in a subsubsection describing plume height.
2. Unclear description about the novelty of the methodology:
   a. In the abstract as well as the main text it is not clear where the novelty is. Is the novel methodology the SOF method? Or is it the plume height estimation? Or is it the expanded method to estimate uncertainty? I suggest reducing the frequency of words such as "novel", "first-time" or "first" and only using in conjunction with the introduction of what is new to the science community.
   b. While the SOF method is not widely used up to this time, it is not novel.
   c. Stationary ground-based applications using FTIR are widespread and well published. Thus, it is generally well known to carefully account for systematic and random spectroscopic uncertainty in the FTIR community.
   d. After several reads through the paper, I believe the sole novelty here is the plume height estimation using the MeFTIR in situ in combination with the SOF columns. This means that only C3 contributes to this novel flux estimation. It is great to have the validation case study and others to further support the range of uncertainty an emission flux has but warrants clearer description within the paper.
   e. What kind of implications do the findings have (Line 26)? In other applications an aircraft or drone *in situ* instrument might measure the vertical distribution of the plume as well as wind speed which would even further constrain flux uncertainty.

3. Similarities to Vechi et al. (2023):
    a. Vechi et al. (2023) only used C3 data. But also already discussed expanded uncertainty and also obtained 37% uncertainty on the flux and used the same LIDAR for wind data. However, plume height estimation used distance between measurement road and source as well as horizontal and vertical winds. I suggest being more clear about the published C3 results in the Vechi et al. (2023) paper and that various information is shown here again but with the difference in plume height estimation.
4. Doublecheck listed uncertainties in Abstract and Results section agree with each other:
    a. The abstract lists the validation test measurement errors from -31 to +14%, estimated expanded uncertainty as 12 to 17% and application to farms as 21 to 37%. However, in the results the expanded uncertainty is listed as 15 to 37% with a median of 27% and Table 2 lists values from 12 to 37%. So I'm not sure if for example the 21% listed in the abstract are correct…

**Minor comments:**

Line 14: Currently reads as if SOF is the novel methodology. Suggest removing "introduces a novel methodology" such that the sentence becomes "This study for evaluating uncertainties in NH3 emissions measurements uses the Solar Occultation Flux (SOF) method."

Line 59ff: The SOF technique has also been demonstrated from aircraft (Kille et al. 2022)

Line 61, 77, 84, 104, and more: Suggest using terms and abbreviations slant column density (SCD) and vertical column density (VCD) instead of "slant columns" or simply "columns". Other literature uses the established terms SCD and VCD for slant and vertical columns as well as air mass factor (AMF) as the term describing their relation (to name a few, see e.g. Eq 1 in Griffin et al. (2021) or Eq 1 and 2 in Kuhlmann et al. (2022))

Line 73: While the measurement column is being driven through the gas plume, I might consider changing the sentence to "driving below the gas plume" as the vehicle and instrument detectors are below the (majority) of the plume.

Line 74: Photons or solar light is captured by the solar tracker and spectra by the spectrometer. Suggested rephrasing to something such as "A solar tracker, containing several mirrors, follows the sun as the car moves and transmits solar light to the spectrometer where spectra are captured during sunny or low cloud coverage conditions." Please note that I suggested changing "follows the light" to "follows the sun" as light could refer to either scattered or direct light.

Line 84: I believe you mean "calculated enhanced column" instead of "calculated column"? The spectrum outside a plume is not necessarily equal to 0 especially if there are upwind contributions.

Line 85: How is that sentence on "low gas concentration should be chosen as the reference" to be understood? The reference or background should be representative of the external condition and not the one point with the lowest measurement.

Line 86: How is the "new reference spectrum" created?

Line 86: What is meant by "absolute column"?

Figure 1b: Is the residual multiplied by a factor? The difference between "Fit" and "Meas" looks negligible at the two peaks but shows visibly in the residual. The colors red and orange are too similar.

Figure 1d: Labels on inset are too small.

Figure 2: Does panel b have the same value range on the y axis? It appears that y = 0 here intersects at the x axis instead of y = -25 as in panel a.

Line 138: At which height above ground did SOF and MeFTIR measure from the vehicle?

Line 145: Should the solar angle not be contained in the integral with the column similar to Eq 1 as the angle changes over time and is somewhat unique to each measurement?

Line 217: You state this is "the first time that uncertainties in NH3 SOF emission measurements from livestock production have been established", but in Line 61 you state that SOF "has been recently used to measure agricultural NH3 emission sources (Kille et al., 2017, Vechi et al., 2023)". Vechi et al., 2023 uses the same data so this is not the first time and Kille et al., 2017 also measured concentrated animal feeding operations. I suggest rewriting this sentence to include the plume height as what has been contributed the first time.

Line 225: It is unclear what is meant by "novel method to assess spectroscopic uncertainties". Spectroscopic uncertainties are well documented in FTIR publications such as from stationary FTIR networks. See for example Table 3 in Viatte et al. (2014).

Line 226: Whereas the previous sentence focused on spectroscopic uncertainty, which is also a measurement, this new paragraph seems to imply that measurement means emission flux measurement. I suggest explicitly stating "Emission flux random uncertainty" instead of "Measurement uncertainty". Additionally, in Line 224 the words "superior results" are chosen when purely assessing spectroscopic uncertainty but in Line 226 (shifting to the emission flux uncertainty) it is stated that wind is the strongest influencer. I suggest modifying the sentences as this immediately seems to weaken the claim that these will be superior results.

Line 238: How is the number 1.96 derived? Wouldn't most of the contribution be from right where the NH3 fingerprints are, so despite your window being smaller than the full band would it not be better to assume the full absorption strength uncertainty?

Line 248ff: In lines 81ff you describe the retrieval window for NH3 to be the broad range from 900-1000 cm-1. Why do you calculate the error for a subregion of the retrieval's spectral range? How do the lines outside this subregion contribute to the retrieval error?

Figure 4a: Be more specific about what uncertainty is portrayed in the panel. Is it the systematic uncertainty? Is column error the least-square sum of random and systematic spectroscopic errors?

Figure 4a: The 1:1 line does not appear dashed except for the last bit whereas the legend indicates dashes.

Figure 4b: Is the residual multiplied by a factor? The difference between "Fit" and "Meas" looks negligible at the two peaks but shows visibly in the residual. The colors red and orange are too similar.

Line 291 and Figure 5: The sentence describes it like Acol is the integrated column area across the plume, but in the figure caption it says the grey-shaded area is the integrated area in Eq 9. Unclear whether you meant the uncertainty is derived from the difference in the two sides' background columns or the area between the columns and temporal/spatial distance of the plume?

Line 336: What is meant by effective degree of freedom? What is meant by "were considered"? Were they applied or found unnecessary?

Line 353: What is "a large number of transects"?

Line 359: Suggest expanding section title to state this is specifically for C3. Otherwise it sounds like MeFTIR and SOF were always operated simultaneously whereas before it was stated that MeFTIR was only used in C3 (Line 204, and Table 1 also shows it only for C3).

Figure 6: Caption for panel b is somewhat confusingly structured. Suggest something like "Examples of plume height calculation using the two methods VCGC (light red bar; error bars correspond to the variation in the plume height calculation – variation in plume height calculation and resulting wind speeds) and PTVS (dark red bar; the error bars correspond to the variation of the HP calculation – variations in wind speed and measured distance)."

Figure 7b: Suggest adding legend for the lines (1:1 and ?)

Line 392ff: Does "lack of vertical wind profile measurements" imply that wind uncertainties should be increased to remain conservative with flux estimates?

Line 399: Was only NH3 released? Could a second, more stable in behavior, gaseous species be released simultaneously to better determine whether or not NH3 deposition or loss is taking place?

Figure 8: Please also add wind arrows for panels c and d.

Line 427 and 430: What does "for the house only" mean?

Line 443: I usually understand a transect to be a straight line. However, since you state the feedlot had an "18 km perimeter" and that it took "an hour to measure one box transect", did you mean one loop/ circle around the box capturing upwind and downwind or did you mean one side of the box?

Line 448: Please add the range of meteorological factors and time of year for each to support your statement about "High fluctuations".

Line 457ff: Suggest adding a little summary as to what is documented in Vechi et al. (2023). How is the EF estimated in Vechi et al. (2023) if in the previous sentence you stated "the number of animals was not known" and hence EFs "could not be established".

Line 472f: See also Rowe et al. (2022), where CO from airborne SOF was compared to the TROPOMI satellite data product.

**Technical comments:**

Line 46: CSDR should be CRDS

Line 48: close-path should be closed-path

Line 49f: It is not clear what "its" refers to in this sentence. Should this be "their" or "the instruments'"?

Line 86f: "it" in "it results in" could be misinterpreted. Does "it" represent absolute column or calculated enhanced column or something else?

Line 105f: Eq 1 includes wind, so I suggest removing the mention of Eq 1 in Line 105 and keeping it only in Line 106.

Line 135: "Fig. Case II" should be "Fig. 3 Case II"

Line 148: Sentence is unclear with comma between "height, distance". Should it be "available height and distance" or something similar?

Line 201: "concentrate animal" should be "concentrated animal"

Line 249: "AVG-abs960-968um" should be "AVG-abs960-968cm-1"

Line 249 and Eq 6: Should the term in the denominator of Eq 6 be the same as "(AVG-abs960-968cm-1)" in Line 249?

Eq 7: What is "abs" in the denominator here? Should it be the same as in Line 249 and Eq 6?

Line 270: Missing a word after "random" in "1000 random such as these were conducted"

Line 361: "compare, this" should be "compare this"

Figure 7: Is the time in the caption local time or UTC?

Line 441: "quantified" should be "quantify"

**References:**

Griffin et al. (2021): Biomass burning nitrogen dioxide emissions derived from space with TROPOMI: methodology and validation. https://doi.org/10.5194/amt-14-7929-2021

Kille et al. (2022): The CU Airborne Solar Occultation Flux Instrument: Performance Evaluation during BB-FLUX. https://doi.org/10.1021/acsearthspacechem.1c00281

Kuhlmann et al. (2022): Mapping the spatial distribution of NO2 with in situ and remote sensing instruments during the Munich NO2 imaging campaign. https://doi.org/10.5194/amt-15-1609-2022

Rowe et al. (2022): Carbon Monoxide in Optically Thick Wildfire Smoke: Evaluating TROPOMI Using CU Airborne SOF Column Observations. https://doi.org/10.1021/acsearthspacechem.2c00048

Viatte et al. (2014): Five years of CO, HCN, C2H6, C2H2, CH3OH, HCOOH and H2CO total columns measured in the Canadian high Arctic. https://doi.org/10.5194/amt-7-1547-2014

---

## Author Comment (AC1)

**Response to reviewers for manuscript**

Dear Senior Editor,

Thank you for the opportunity to revise our manuscript, "An uncertainty methodology for solar occultation flux measurements: ammonia emissions from agriculture." We greatly appreciate the insightful comments from both reviewers, which will enhance the quality and clarity of our paper. In response to Reviewer #1's comments on the structure and novelty of our work, we have further clarified our contributions. Besides the innovative plume height methodology, we also emphasize our advancements in the use of SOF for $NH_3$ quantification, the application of the GUM methodology for uncertainty estimation particularly and blind validation tests. These aspects collectively underscore the novelty and significance of our research in this field. Reviewer 2# had fewer comments, with the emphasis on the deposition of $NH_3$, which we explained that it diverges from the goal of the paper, to dive deeper in the deposition impact on the quantification. Regarding Reviewer #2's focus on $NH_3$ deposition, we have added a brief discussion to clarify that while deposition impacts are outside the primary scope of our paper, our findings indirectly inform this area. This addition aims to acknowledge the relevance of deposition in the broader context of $NH_3$ quantification without diverting from the main objectives of our study. As feedback from the reviewer #2, the title of the manuscript was changed to "An uncertainty methodology for solar occultation flux measurements: ammonia emissions from livestock production."

We look forward to your decision.

Best Regards,

Johan Mellqvist, Nathalia Thygesen Vechi, and co-authors

| Reviewer Comment | Author' Response | Revised Text – Line numbers refer to clean (without track changes) version of the revised paper. |
|---|---|---|
| | **Reviewer #1** | |
| The authors give a detailed description of the issue of measuring ammonia (NH$_3$) emissions from the agricultural sector. Using the solar occultation flux (SOF) method, which is a remote sensing technique with mobile FTIR, the conducted measurements of NH$_3$ from a validation experiment and three case studies are used in combination with different combinations of wind measurements and an in situ FTIR to infer NH$_3$ emission fluxes and their uncertainties. The paper is appropriate for publication in AMT because it provides a new methodology to infer plume height prior to estimating emission fluxes. However, the specific comments in the next section need to be addressed by the authors beforehand | Thank you for the comments and feedback. | |
| In section 2 the authors introduce a validation and three case studies, but then in the results section are not very specific about the order of how these were introduced. This leads to confusion when e.g. in section 4 it becomes less clear that MeFTIR was only used during C3. It is also confusing when in the results section suddenly specific plume transects are discussed but it is not clear which case study these relate to. I suggest restructuring section 2 and better introducing the measurements | **Suggestion Implemented** In the results section, we first focused on the uncertainty estimation, and therefore the plume height estimation appears before presenting the cases. This section is also in the beginning of the results, because it is one of the highlights of the paper, and therefore it is better if positioned in the beginning than in the end. Additionally, we followed the reviewer comments by having separated sections for | **Added:** Section 2.1.2 Mobile extractive FTIR (MeFTIR) instrument In this section we introduce the MeFTIR instrument inside of the section that explains the plume height calculation, as this is where this instrument will be used. Section 2.3.1 Wind measurements at the Case studies We added a subsection focusing on the wind measurements instruments used at the different case studies. |

| | | |
|---|---|---|
| and studies. It would help to have a designated subsection for the wind related instruments as well as one for the MeFTIR such that it doesn't first appear in a sub-subsection describing plume height. | the MeFTIR and for the instruments used at the different wind measurements. | |
| In the abstract as well as the main text it is not clear where the novelty is. Is the novel methodology the SOF method? Or is it the plume height estimation? Or is it the expanded method to estimate uncertainty? I suggest reducing the frequency of words such as "novel", "first-time" or "first" and only using in conjunction with the introduction of what is new to the science community. | **Clarification provided**

We rephrased in a few parts in the text. The novelty of this paper is threefold.

We also remove the words "novel" and "first-time". | **Added Line 73:** The novelty in the paper is threefold: (1) The plume height methodology, (2) validation of $NH_3$ measurements by SOF and (3) the uncertainty calculation following the Guide to the Expression of Uncertainty in Measurement (GUM) methodology. |
| While the SOF method has not been widely used up to this time, it is not novel. | **Clarification provided**

That is true, we also mention this in the text, "The Solar occultation flux (SOF) has been used for years (Line 60)". The novelty, as mentioned before, lies in another points, we tried to clarify this throughout the text. | **No modification** |
| Stationary ground-based applications using FTIR are widespread and well published. Thus, it is generally well known to carefully account for systematic and random spectroscopic uncertainty in the FTIR community. | **Clarification provided**

The systematic uncertainties in SOF measurements of $NH_3$ from farms includes solar spectroscopy and gas flux calculations based on mobile measurements and wind speed, differing fundamentally from ground concentration measurements, and introducing specific uncertainties for farm emissions. The error analysis for spectroscopy in our study was conducted in a novel manner. To ensure the accuracy of this approach, we employed spectroscopic | **No modification** |

| | | |
|---|---|---|
| | simulations, comparing them with commonly used uncertainty calculations. This comparison demonstrated that our method yields more precise uncertainty estimates. | |
| After several reads through the paper, I believe the sole novelty here is the plume height estimation using the MeFTIR in situ in combination with the SOF columns. This means that only C3 contributes to this novel flux estimation. It is great to have the validation case study and others to further support the range of uncertainty an emission flux has but warrants clearer description within the paper. | **Clarification provided**

Motivation of the paper is wider than what the reviewer see as we point out in the paper: The novelty in the paper is threefold: (1) The plume height description, (2) the validation of the SOF method for $NH_3$ gas, and (3) the uncertainty methodology for SOF following the GUM methodology. The latter includes a new way of describing spectroscopic uncertainty. | **Added Line 73:** The novelty in the paper is threefold: (1) The plume height description, (2) the validation of the SOF method for $NH_3$ gas, and (3) the uncertainty methodology following the Guide to the Expression of Uncertainty in Measurement (GUM) methodology. |
| What kind of implications do the findings have (Line 26)? In other applications an aircraft or drone *in situ* instrument might measure the vertical distribution of the plume as well as wind speed which would even further constrain flux uncertainty. | **Clarification provided**

The plume height method can be used in measurements of several other sources using SOF for measurements of VOCs from oil and gas sector. The uncertainty estimates are, with some modification, also valid for other sources and when using other similar methods, such as mobile DOAS. Additionally, we also showed the application in different sizes of livestock production, and that this method can be further used for measure NH3 from other agricultural sources as fertilized fields. | **Rephrased Line 25:** This paper's findings offer potential for broader applications, such as measuring $NH_3$ fluxes from fertilized fields, as well as in the oil and gas sector. However, these applications would require further research to adapt and refine the methodologies for these specific contexts. |
| Vechi et al. (2023) only used C3 data. But also already discussed expanded uncertainty and also obtained 37% uncertainty on the flux and used the same LIDAR for wind data. However, | **Clarification provided:**

It is true that there are some connections between both papers, although they were published in an unintended order. Ideally, this paper should have been released first, as it | **Added:**

**Line 64:** The present paper is focused on the methodology and uncertainties of $NH_3$ measurements using SOF, while in the previously published Vechi et al. (2023) the attention is on the |

| | | |
|---|---|---|
| plume height estimation used distance between measurement road and source as well as horizontal and vertical winds. I suggest being more clear about the published C3 results in the Vechi et al. (2023) paper and that various information is shown here again but with the difference in plume height estimation. | complements the other paper (Vechi et al., 2023).

The key difference lies in the content: this paper provides a detailed analysis of the SOF (Solar Occultation Flux) uncertainty with examples of different systematic uncertainties. In contrast, Vechi et al., (2023) only briefly discusses the uncertainty calculation. Additionally, the data used in this paper (C3) was obtained during the same campaign, but it was not included in Vechi et al., 2023. This paper further includes a validation exercise and the plume height methodology.

While the other paper focuses on interpreting emission data using combined methods (MeFTIR and SOF), this paper concentrates on NH3 measurements from sources, focusing on different applications (smaller sources, pigs, cows).

Importantly, it is not necessary for the reader to have read Vechi et al. 2023 to understand the content of this paper | results from measurements using this methodology, therefore they are supplementary to each other.

**Line 482:** In Vechi et al. (2023), similar measurements of $NH_3$ by SOF was performed, and EF's were calculated according to the number of animals provided by San Joaquin air quality district, additionally a diurnal pattern was observed, with emissions being highest around 12:00. |
| The abstract lists the validation test measurement errors from -31 to +14%, estimated expanded uncertainty as 12 to 17% and application to farms as 21 to 37%. However, in the results the expanded uncertainty is listed as 15 to 37% with a median of 27% and Table 2 lists values from 12 to 37%. So I'm not sure if for example the 21% listed in the abstract are correct… | **Clarification provided**

The discrepancy arises because in the validation results, we calculated an average of the uncertainties mentioned in the results section but did not do so in the abstract and Table 2. We have added a footnote to Table 2 explaining the origin of the 15% figure. The 21% figure is related to the examples from the farm measurements | **Added to table 2:**

Average 15% added to table 2 and footnote.

… [1]Average of the uncertainties found in the validation study. |

| | | |
|---|---|---|
| Currently reads as if SOF is the novel methodology. Suggest removing "introduces a novel methodology" such that the sentence becomes "This study for evaluating uncertainties in NH3 emissions measurements uses the Solar Occultation Flux (SOF) method." | **Suggestions implemented**

The novelty is the uncertainty as the sentence follows … "methodology for evaluating uncertainties in $NH_3$". We did however implemented the reviewer suggestions to make the text more clear. | **Rephrased Line 14**: This study presents methodology for the estimation of uncertainties in … |
| The SOF technique has also been demonstrated from aircraft (Kille et al. 2022) | **Suggestion Implemented**

Reference was added | **Added Line 62:** SOF has been used on a mobile platform and in an aircraft (Kille et al., 2022). |
| Line 61, 77, 84, 104, and more: Suggest using terms and abbreviations slant column density (SCD) and vertical column density (VCD) instead of "slant columns" or simply "columns". Other literature uses the established terms SCD and VCD for slant and vertical columns as well as air mass factor (AMF) as the term describing their relation (to name a few, see e.g. Eq 1 in Griffin et al. (2021) or Eq 1 and 2 in Kuhlmann et al. (2022)) | **Clarification provided**

We maintain the use of 'columns' in place of 'slant column density' (SCD), as these terms are synonymous. This aligns with the terminology used in many scientific papers, including our own works such as Vecchi 2023, Mellqvist 2010, and Johansson 2014 | **No modification** |
| Line 73: While the measurement column is being driven through the gas plume, I might consider changing the sentence to "driving below the gas plume" as the vehicle and instrument detectors are below the (majority) of the plume. | **Suggestion implemented**

Suggestion was implemented according to the suggestion. | **Rephrased Line 80**: … spectra while driving below the gas plume … |
| Line 74: Photons or solar light is captured by the solar tracker and spectra by the spectrometer. Suggested rephrasing to something such as "A solar tracker, containing several mirrors, follows the sun as the car moves and transmits solar light to the spectrometer | **Suggestion implemented**

Suggestion was implemented according to the suggestion. | **Rephrased Line 81:** … A solar tracker, containing several mirrors, follows the sun as the car moves and transmits solar … |

| | | |
|---|---|---|
| where spectra are captured during sunny or low cloud coverage conditions." Please note that I suggested changing "follows the light" to "follows the sun" as light could refer to either scattered or direct light. | | |
| Line 84: I believe you mean "calculated enhanced column" instead of "calculated column"? The spectrum outside a plume is not necessarily equal to 0 especially if there are upwind contributions | **Suggestion implemented**

Suggestion was implemented according to the suggestion | **Rephrased line 91:** The calculated enhanced column values … |
| Line 85: How is that sentence on "low gas concentration should be chosen as the reference" to be understood?
The reference or background should be representative of the external condition and not the one point with the lowest measurement. | **Suggestion implemented**

Suggestion was implemented according to the suggestion. | **Rephrased line 92:** Ideally, a location a representative of the external conditions should be chosen as the reference. |
| Line 86: What is meant by "absolute column"? | **Clarification provided**

The spectral retrieval is done by rationing all spectra with a reference spectrum, recorded outside the plume. In this way the retrieved values are all relative to the value in the reference spectrum and they are not absolute. | **Rephrased line 93:** While retrieval of absolute columns is possible, which is without decreasing the reference, however the column results in … |
| Figure 1b: Is the residual multiplied by a factor? The difference between "Fit" and "Meas" looks negligible at the two peaks but shows visibly in the residual. The colors red and orange are too similar. | **Clarification provided**

The information is indeed correct. Please note that the values on the y-axis are relatively low, which might have contributed to the difficulty in discerning differences, especially due to the similar colours used. We have double-checked this and can confirm its accuracy. | **Figure 1:** Colours were changed for better visualization. |
| Figure 1d: Labels on inset are too small. | **Suggestion implemented**

Figured was made a bit larger | **Figure 1 - modified** |

| | | |
|---|---|---|
| Figure 2: Does panel b have the same value range on the y axis? It appears that y = 0 here intersects at the x axis instead of y = -25 as in panel a. | **Suggestion implemented**

Figure was corrected | **Figure 2 - modified** |
| Line 138: At which height above ground did SOF and MeFTIR measure from the vehicle? | **Suggestion implemented**

Approximately 2 meters**.** | **Added line 114:** The MeFTIR was sampling from the top of the car's roof, at about 2 m from the ground, while the SOF mirrors were also positioned at approximately the same distance. |
| Line 145: Should the solar angle not be contained in the integral with the column similar to Eq 1 as the angle changes over time and is somewhat unique to each measurement? | **Suggestion implemented**

We can move the cos to inside the eq. 3. However, this is only in the case you make a long transect, which was not applicable in any of the examples used. | **Eq. 3 modified** |
| Line 217: You state this is "the first time that uncertainties in NH3 SOF emission measurements from livestock production have been established", but in Line 61 you state that SOF "has been recently used to measure agricultural NH3 emission sources (Kille et al., 2017, Vechi et al., 2023)". Vechi et al., 2023 uses the same data so this is not the first time and Kille et al., 2017 also measured concentrated animal feeding operations. I suggest rewriting this sentence to include the plume height as what has been contributed the first time. | **Suggestion implemented**

According to the reviewer suggestion

The methodology in this paper adheres to the GUM (Guide to the Expression of Uncertainty in Measurement) approach, type A. It involves calculating uncertainty for each individual measurement, considering flux variability, other measured factors, and systematic uncertainties. This approach is widely adopted by meteorological institutes globally.

The Vecchi 2023 paper utilizes the outcomes of these calculations but omits the specifics, which are provided here. It's important to note that these papers were published out of sequence, with the latter simply utilizing the findings of the former.

Additionally, we introduce a novel method for calculating systematic spectroscopic uncertainty and clearly define confidence intervals. | **Rephrased line 236:** … This shows for the first time the uncertainties in $NH_3$ SOF emission measurements from livestock production based on the GUM approach ((Joint Committee For Guides In Metrology, 2008), and shows for the first time the methodology for plume height calculation, albeit drawing ... |

| | | |
|---|---|---|
| | Contrastingly, the Kille 2017 paper calculates uncertainties differently. They do not use measured but modelled winds and can therefore not compute random uncertainty for each measurement as per GUM principles, thus individual uncertainties for each measurement are not ascertainable.

. | |
| Line 225: It is unclear what is meant by "novel method to assess spectroscopic uncertainties". Spectroscopic uncertainties are well documented in FTIR publications such as from stationary FTIR networks. See for example Table 3 in Viatte et al. (2014). | **Clarification provided**

This section outlines the systematic uncertainty arising from imperfect spectroscopic fitting of band shapes. Typically, the Root Mean Square (RMS) of the residuals from the fit is used to assess this directly at a single frequency point. However, in this paper, we employ multiple rovibrational lines to retrieve NH3. Since multiple lines are used, they should reduce the uncertainty compared to measuring a single line. In our new model, the uncertainty ($U_{rel,1}$) decreases proportionally with the square root of the number of samples, analogous to sampling error. To validate this approach, we conducted a simulation of the spectroscopic error and then tested three different methods, as described in the paper ($U_{rel,1}$, $U_{rel,2}$, and $U_{rel,3}$). Although not describing the uncertainty perfectly, this method greatly improved the description of the spectroscopic uncertainty compared to more conventional ways using direct RMS ($U_{rel,2}$) or an average of RMS ($U_{rel,3}$). | **No modification** |
| Line 226: Whereas the previous sentence focused on spectroscopic uncertainty, which is also a measurement, this new paragraph | **Suggestion implemented**

According to the reviewer suggestions | **Rephrased line 246:** Emissions measurement random uncertainty … |

| | | |
|---|---|---|
| seems to imply that measurement means emission flux measurement. I suggest explicitly stating "Emission flux random uncertainty" instead of "Measurement uncertainty". | | |
| Additionally, in Line 224 the words "superior results" are chosen when purely assessing spectroscopic uncertainty but in Line 226 (shifting to the emission flux uncertainty) it is stated that wind is the strongest influencer. I suggest modifying the sentences as this immediately seems to weaken the claim that these will be superior results. | **Clarification provided**

 The term 'superior results' refers to the effectiveness of the new method (Uret,1) in handling spectroscopy errors, in comparison to the other methods (Uret,2 and Uret,3). | **Rephrased line 243**: As part of the uncertainty, description this study proposes a new method to assess spectroscopic uncertainties, demonstrating superior results to improved spectroscopy uncertainties when compared to the approach typically used in general spectroscopic measurements. |
| Line 238: How is the number 1.96 derived? Wouldn't most of the contribution be from right where the NH3 fingerprints are, so despite your window being smaller than the full band would it not be better to assume the full absorption strength uncertainty? | **Clarification provided**

 Following the GUM (Guide to the Expression of Uncertainty in Measurement) procedure, the relative 1s uncertainties are added in quadrature, and then the square root of the sum is taken. Ultimately, this sum is multiplied by the coverage factor, denoted as k. We assume that all systematic uncertainties are accompanied by a specific uncertainty distribution. Here, we presume this distribution is characterized by a 65% confidence limit, which typically corresponds to a factor of 1.96. | **No modification** |
| Line 248ff: In lines 81ff you describe the retrieval window for NH3 to be the broad range from 900-1000 cm-1. Why do you calculate the error for a subregion of the retrieval's spectral range? How do the lines outside this subregion contribute to the retrieval error? | **Clarification provided**

 To assess the retrieval error (Uret), we calculate the ratio of the average NH3 absorbance in the 960 to 968 cm^-1 range to the standard deviation of the fitting residual (STD) within the same wavelength range. This ratio is then divided by the square root of the number of points. | **No modification** |

| | | |
|---|---|---|
| | The rationale for focusing exclusively on this interval is that it is where the primary NH3 information content is derived from. If we were to consider the full band, the RMS (Root Mean Square) would likely be influenced by other interfering species, particularly water. This is because $NH_3$ lines are only present in specific parts of the window, and a full-band RMS would not accurately indicate whether there are systematic issues specifically with the fitting of $NH_3$. | |
| Figure 4a: Be more specific about what uncertainty is portrayed in the panel. Is it the systematic uncertainty? Is column error the least-square sum of random and systematic spectroscopic errors? | **Suggestion implemented**

We added in the legend that this is correspondent to systematic uncertainty. | **Added to the figure 4a legend:** Column errors and systematic uncertainty. |
| Figure 4a: The 1:1 line does not appear dashed except for the last bit whereas the legend indicates dashes. | **Suggestion implemented**

Figure was corrected | **Figure 4 - modified** |
| Figure 4b: Is the residual multiplied by a factor? The difference between "Fit" and "Meas" looks negligible at the two peaks but shows visibly in the residual. The colors red and orange are too similar. | **Clarification provided**

As previously mentioned, the information is indeed correct. Please note that the values on the y-axis are relatively low, which might have contributed to the difficulty in discerning differences, especially due to the similar colors used. We have double-checked this and can confirm its accuracy. | **No modification.** |
| Line 291 and Figure 5: The sentence describes it like Acol is the integrated column area across the plume, but in the figure caption it says the grey-shaded area is the integrated area in Eq 9. Unclear whether you meant the uncertainty is derived from the difference in the two sides' background columns or the area between the | **Clarification provided and suggestion implemented.**

Yes, there is an error in Equation 9 due to differing units between Acol and $\Box$col. We should integrate across the plume in the | **Eq 9** modified to

$$U_B = \frac{\frac{\int_{l1}^{l2} \Delta col_{Background}\ dl}{2}}{\sqrt{3} \cdot A_{col}}$$ |

| columns and temporal/spatial distance of the plume? | same manner as Acol. This has been done this in the calculation of the uncertainty | |
|---|---|---|
| Line 336: What is meant by effective degree of freedom? What is meant by "were considered"? Were they applied or found unnecessary? | **Clarification provided**

In the GUM (Guide to the Expression of Uncertainty in Measurement) methodology, the effective degrees of freedom is defined as the average degrees of freedom across all parameters. The specific formula used for this calculation is the Welch-Satterthwaite equation. This equation is particularly important when estimating the combined standard uncertainty, especially in cases where individual components of uncertainty are estimated with different degrees of freedom. | **Rephrased Line 357:**

Furthermore, by considering the methodology, the effective degrees |
| Line 353: What is "a large number of transects"? | **Clarification provided**

We have made changes. | **Rephrased line 374:**

… Moreover, for most case studies, several transects was recorded (>5) … |
| Line 359: Suggest expanding section title to state this is specifically for C3. Otherwise it sounds like MeFTIR and SOF were always operated simultaneously whereas before it was stated that MeFTIR was only used in C3 (Line 204, and Table 1 also shows it only for C3). | **Suggestion implemented**

We acknowledge that the structure of the plume height description is somewhat complex. However, we chose to maintain its placement in the manuscript to emphasize its importance, even before Campaign 3 (C3) is discussed. Nonetheless, we have rephrased the title as suggested by the reviewer. | **Title was changed to:**

4.1.1. Plume height ($H_P$) in case study 3. |
| Figure 6: Caption for panel b is somewhat confusingly structured. Suggest something like "Examples of plume height calculation using the two methods VCGC (light red bar; error bars correspond to the variation in the plume height calculation – variation in plume | **Suggestion implemented**

Legend was rephrased according to the reviewer suggestions. | **Figure 6 legend:**

Examples of plume height calculation using the two methods (VCGC), (light red bar; error bars correspond to the variation in the plume height calculation – corresponding to variation on |

| | | |
|---|---|---|
| height calculation and resulting wind speeds) and PTVS (dark red bar; the error bars correspond to the variation of the HP calculation – variations in wind speed and measured distance)." | | distances and wind speed) and (PTVS) (dark red bar; the error bars correspond to the variation of the $H_P$ calculation). |
| Figure 7b: Suggest adding legend for the lines (1:1 and ?) | **Suggestion implemented**

Implemented according to the reviewer suggestions | **Figure 7 modified** |
| Line 392ff: Does "lack of vertical wind profile measurements" imply that wind uncertainties should be increased to remain conservative with flux estimates? | **Clarification provided**

We assessed the wind uncertainties separately for each campaign based available wind information. Yes, they would increase of there is limited wind information. | **No modification** |
| Line 399: Was only NH3 released? Could a second, more stable in behavior, gaseous species be released simultaneously to better determine whether or not NH3 deposition or loss is taking place? | **Not implemented**

This is a good idea, however we did not release any other gas that could be measured by the SOF in the experiment | **No modification** |
| Figure 8: Please also add wind arrows for panels c and d. | **Suggestion implemented**

They are actually there, but we made them bigger. | **Figure 8 modified** |
| Line 427 and 430: What does "for the house only" mean? | **Suggestion implemented**

Clarification was added. It means without accounting for the manure tank (Fig. 4b). | **Added line 449:** … house only, not accounting for the manure tank ... |
| Line 443: I usually understand a transect to be a straight line. However, since you state the feedlot had an "18 km perimeter" and that it took "an hour to measure one box transect", did you mean one loop/ circle around the box capturing upwind and downwind or did you mean one side of the box? | **Clarification provided**

A Box transect means that we drive in a circle around the farm to capture both upwind and downwind. (Fig. 4a). Also this is specified in the text … an hour to measure one box transect ... | **No modification** |

| | | |
|---|---|---|
| Line 448: Please add the range of meteorological factors and time of year for each to support your statement about "High fluctuations". | **Suggestion implemented**

Yes, the information was added accordingly. | **Added line 469:**

4.3 g head$^{-1}$ h$^{-1}$ (Annual emission 2015) ranging from 1.1 – 51 g head$^{-1}$ h$^{-1}$. In contrast, other studies showed larger Efs as 18.5 to 42 g head$^{-1}$ h$^{-1}$ (October 2014 and June 2015)(Leifer et al., 2017, 2018) and 14.9 to 79.7 g head$^{-1}$ h$^{-1}$ (May and June 2010)(Nowak et al., 2012) |
| Line 457ff: Suggest adding a little summary as to what is documented in Vechi et al. (2023). How is the EF estimated in Vechi et al. (2023) if in the previous sentence you stated "the number of animals was not known" and hence EFs "could not be established". | **Suggestion implemented**

Suggestion implemented according to the reviewer suggestion. | **Added line 482:** In Vechi et al. (2023), similar measurements of NH$_3$ by SOF was performed, and EFs were calculated according to the number of animals provided by San Joaquin air quality district, additionally a diurnal pattern was observed, with emissions being highest around 12:00. |
| Line 472f: See also Rowe et al. (2022), where CO from airborne SOF was compared to the TROPOMI satellite data product. | **Suggestion implemented**

Reference was added to the paper. | **Added line 498**: … and also for CO measurements by SOF (Rowe et al., 2022). |
| Technical comments

Line 46: CSDR should be CRDS
Line 48: close-path should be closed-path
Line 49f: It is not clear what "its" refers to in this sentence. Should this be "their" or "the instruments'"?
Line 86f: "it" in "it results in" could be misinterpreted. Does "it" represent absolute column or calculated enhanced column or something else?
Line 105f: Eq 1 includes wind, so I suggest removing the mention of Eq 1 in Line 105 and keeping it only in Line 106.
Line 135: "Fig. Case II" should be "Fig. 3 Case II"
Line 148: Sentence is unclear with comma between "height, distance". | **Suggestions implemented**

Suggestions were implemented according to the reviewer suggestions. | **Corrections made according to the reviewer suggestion.** |

| | | |
|---|---|---|
| Should it be "available height and distance" or something similar?
Line 201: "concentrate animal" should be "concentrated animal"
Line 249: "AVG-abs960-968um" should be "AVG-abs960-968cm-1"
Line 249 and Eq 6: Should the term in the denominator of Eq 6 be the same as "(AVG-abs960-968cm$^{-1}$)" in Line 249?
Eq 7: What is "abs" in the denominator here? Should it be the same as in Line 249 and Eq 6?
Line 270: Missing a word after "random" in "1000 random such as these were conducted"
Line 361: "compare, this" should be "compare this"
Figure 7: Is the time in the caption local time or UTC?
Line 441: "quantified" should be "quantify" | | |

---

## Author Comment (AC2)

**Response to reviewers for manuscript**

Dear Senior Editor,

Thank you for the opportunity to revise our manuscript, "An uncertainty methodology for solar occultation flux measurements: ammonia emissions from agriculture." We greatly appreciate the insightful comments from both reviewers, which will enhance the quality and clarity of our paper. In response to Reviewer #1's comments on the structure and novelty of our work, we have further clarified our contributions. Besides the innovative plume height methodology, we also emphasize our advancements in the use of SOF for $NH_3$ quantification, the application of the GUM methodology for uncertainty estimation particularly and blind validation tests. These aspects collectively underscore the novelty and significance of our research in this field. Reviewer 2# had fewer comments, with the emphasis on the deposition of $NH_3$, which we explained that it diverges from the goal of the paper, to dive deeper in the deposition impact on the quantification. Regarding Reviewer #2's focus on $NH_3$ deposition, we have added a brief discussion to clarify that while deposition impacts are outside the primary scope of our paper, our findings indirectly inform this area. This addition aims to acknowledge the relevance of deposition in the broader context of $NH_3$ quantification without diverting from the main objectives of our study. As feedback from the reviewer #2, the title of the manuscript was changed to "An uncertainty methodology for solar occultation flux measurements: ammonia emissions from livestock production."

We look forward to your decision.

Best Regards,

Johan Mellqvist, Nathalia Thygesen Vechi, and co-authors

| Reviewer Comment | Author' Response | Revised Text – Line numbers refer to clean (without track changes) version of the revised paper. |
|---|---|---|
| | **Reviewer #2** | |
| The authors address a comprehensive approach to quality uncertainties of measured ammonia emissions from livestock sources by using SOF & MeFTIR instruments, which is well fit for the scope of AMT. Meanwhile, scientifically speaking, understanding the level of uncertainty is crucial for the reliability and validity of the results in the Nitrogen domain.

The authors validated such error estimation method using tracer release experiment (single point source) and later also applied it into multiple livestock farms case studies (area sources), which can further help mitigate the possible causes of errors in real-world. In all, it is worthy of scientific publication. But before that, I do have some specific comments related to the current error propagation methodology and the structural of the paper as well, which I'd hope the authors can remedy certain issues and deepen the discussion. | **Thank you for the comments and feedback** | |
| **Some specific comments:**

1. It is well known that ammonia flux has its bidirectional character. Although notably the authors stated it is out of scope for the present work, I'd like to know how problematic would it be to fully | **Clarification provided**

The paper was focused on presenting the method. As well as the error was focused on the methodology error, so it would not be correct to add in the error budget.

We could have tried to estimate, but it would vary a lot from case by case, so it would | **Added line 506:**

Regarding $NH_3$ deposition, it will very largely according to the conditions of each site. In California for example, where both case studies 2 and 3 were performed, previous studies measured a deposition of 15% in the first 3 km, while others estimated to be from 8 – 15 % (Miller et al., 2014). For Denmark and France, these number might be higher because there was likely less convection during the measurement days, |

| | | |
|---|---|---|
| ignore it from the error propagation? How large will it affect the error ranges? For example, the authors directly compared the tracer emitted NH$_3$ value with the SOF measured value ~200 meter away from the source at the downwind direction. This possibly causes the current error propagation scheme (methodology) to tentatively overestimate the actual 'error', because the ammonia 'deposition loss' is not corrected before the error propagation starts. In the other word, it might not be fair to name ammonia deposition loss as part of the total measurement error, isn't it? | need a whole methodology to estimate that in addition to what we already had.

We added some extra information on this in the last section. | but this paper focused more in describing SOF, rather than investigating the emissions sources. |
| 2. Some comprehensive discussions are missing from the current paper structure. I'd think it can add some extra for this paper if the authors can properly address the above mentioned issue, by adding some comparison cases studies, or discuss both limitations and advantages of the current method and results in the structural way. | **Not implemented**

We are uncertain about the specific point the reviewer is addressing. Our understanding is that it may be an extension of the question regarding deposition. Consequently, we refer to our response to that question and the additional text provided in the previous comment for clarification. | **No modification** |
| 3. Both in the abstract and conclusion parts, the authors emphasized the SOF method estimated NH$_3$ emission can be as low as 1 kg/h ± 21%. Does it somehow indicate the SOF method no longer trustworthy if the | **Clarification provided**

It is most likely the quantification limit from the method, considering the precision of the instrument. It might work for lower emissions at certain meteorological conditions. We added the 0.5 information in the abstract, | **Added line 23:**

(~0.5 - 1 kg h$^{-1}$) and |

| | | |
|---|---|---|
| ammonia source emission below 1 kg/h? As it can influence the applications for potential users, can the authors justify your statement? Is such statement applicable for all agricultural sources or need further testing? | because we believe that even at the 0.5 kg/h the measurements had good quality. | |
| Some specific technical comments are listed below:

1. The title of this paper is 'An uncertainty methodology for solar occultation flux measurements: ammonia emissions from agriculture'. It seems too broad to cover all agricultural domain. The agricultural activities include fertilizer application, livestock operations, and other agricultural processes as well. However, none of the study cases listed in the present work demonstrated its application for fertilizer or manure field emission measurement yet. All cases are focused on livestock farms. Would it be more appropriate  to narrow down the title to 'livestock' ? | **Suggestion implemented**

We agree we the reviewer suggestion, the title should be changed. | **Paper's title was modified** |
| 1. In Line 19, how does the plume height estimation reducing the measurement uncertainty? I could not find detailed discussion in the main text. please reconsider its | **Clarification provided**

We have not calculated the effect explicitly but used the height for our wind calculations. | **No modifications** |

| | | |
|---|---|---|
| value when it is mentioned in the abstract? | | |
| 2. The last sentence in the abstract should be removed. The main paper does not provide concrete evidence or results demonstrating the applicability of the methodology to other gaseous species or purposes, it would not be appropriate to make such claims in the abstract. The abstract should accurately summarize the scope and results of the study based on what is presented in the main paper. It's better to phrase it as a potential avenue for future research or exploration rather than making definitive statements. Mentioning the potential for broader applications in the discussion section of the paper, along with the need for further research in those areas, would be more appropriate. | **Suggestion implemented**

The sentence was changed | **Rephrased Line 25:** This paper's findings offer potential for broader applications, such as measuring $NH_3$ fluxes from fertilized fields, as well as in the oil and gas sector. However, these applications would require further research to adapt and refine the methodologies for these specific contexts. |
| 3. The caption of Figure 2 mentioned case study C3 out of blue without any other context in the previous text, please add some extra information or consider reorder the main text structure. | **Clarification provided**

We have removed the information that it was from case 3 | **Rephrased figure 2 legend**: a) Example of wind profiles, the grey lines show … |
| 4. In line 170, both 2D sonic anemometer and vane wind meter are used in multiple campaigns. How do different | **Clarification provided**

The measurements error of the wind sensor were not taken into account since they | **No modification** |

| | | | |
|---|---|---|---|
| | types of wind meters contribute to the final measurement error? Do you take the instrument system error into account when estimating the wind profile error? | were considered small compared to the absolute wind uncertainty. | |
| 5. | In line 238, absorption strength uncertainty of 2% was assigned from a previous study, is it always a fixed value for all SOF instrument in various application? If not so, can it be properly estimated or not? | **Clarification provided** This is based on the cited reference according to the text and is applied in all the instruments using HITRAN NH3 absorption cross sections. | **No modification** |
| 6. | In line 255 equation 7, the 'abs' value is used. Can you please define what is it? As in equation 6 there is another abs(960-968). Are they the same? Similarly, in equation 8, it is Aabs. They become bit confusing. | **Clarification provided** There was ambiguity in the accompanying text and in equations 6 and 7; this has been fixed. | **Modifications:** Definition of $abs_{avg}$ has been corrected in the text (line 269) Terminology has been changed in Eq.6 to $abs_{avg}$ Terminology has been changed in Eq.7 to $abs_{avg}$ |
| 7. | Some confusing statement occurred in both line 304 and 308. From the previous context, in line 304 C2 should be C1; and in line 308 C1 in fact refers to C2. Please double check. | **Suggestion implemented** The reviewer is correct, the suggestion was implemented. | **Rephrased line 324:** In the validation test and C1, two wind masts … **Line 328:** For C2, only one 10 m mast was used … |
| 8. | In line 317 Table 1. Case study C2 integrated wind profile is estimated using C3 data. Can you explain why C2 can use C3 wind profile data? I assume they were not measured in the same day nor the | **Clarification provided** The campaigns were performed in nearby area and same month but different year. The flux calculation was not performed with the C3 data, instead, information from C3 campaign was used for the uncertainty | **Rephrased line 328:** For C2, only one 10 m mast was used to measure the wind, so we estimated the error of choosing different vertical profiles by using information from another study at the same geographic location and at similar time of the year, because of the lack of data to estimate the real wind profile … |

| | | |
|---|---|---|
| same location, so why do they share the same IWPavg value? | calculation in the estimation of the wind speed average profile. | |
| 9. Line 360, MeFTIR and SOF both used to estimate plume height. However, the current method did not mention MeFTIR measurement uncertainty at the ground level. How large uncertainty can be generated from MeFTIR system? and how large can it contribute to the total error prorogation? | **Clarification provided**

The uncertainty of the MeFTIR concentration measurements is detailed in Vecchi 2023, typically ranging between 5-10%. We have not estimated the uncertainty of the plume height estimation in this paper, and therefore, the MeFTIR concentration uncertainty is not relevant here. Note that we only used the standard deviation of the individual plume height measurements as a measure of their variability, as illustrated in Figure 6. | **No modification** |
| 10. Line 367, the authors stated PTVS method result is slightly lower than VCGS method, but they are similar". I would not agree so. In Figure 6b, it clearly shows PTVS results can be 2 times bigger than VCGC method in many cases. | **Clarification provided**

We phrased the sentence so instead we used the calculated difference between the two methods. Highlighting that small difference might not large interfere in the calculation, especially if they are at higher altitudes. The differences might be also due to the some of the plumes not have an homogenous mixing. | **Rephrased line 387:** … was utilized and compared to the VCGC method, showing that the first method produced in average emissions 35% higher than the second, where only one of the farms (farm 8) had a large difference. |
| 11. Figure 6b, what do you mean "farms" in the title of x-axis? It never mentioned in the main text. Or do you mean "transect" number from certain case study ? please clarify. | **Clarification provided**

We added this information in the figure legend. It means that we used the data from SOF measurements at nine different farms to illustrate the plume height methodology. Because the measurements were made at similar conditions (wind speed and distance) the expected plume height at all the respective farms measurements would expect to be similar. | **Rephrased in figure 6 legend**: b) Examples of average plume height calculation from measurements at nine farms using … |
| 12. Line 383, Figure 7b and line 403 all related to the Sep 22 | | **No modification** |

| | | |
|---|---|---|
| measurement result, which was measured in a cloudy day. Does such data point still validate?  Firstly, it is conflict with the previous statement that SOF is better used in sunny and less cloudy day. Secondly, even without 'deposition loss' correction, SOF measured downwind value is higher than the actual tracer release value. This strongly suggests such data may be too faulty to be trusted. | **Clarification provided**

High-level clouds and haze can lead to greater measurement variability, but all measurements taken on September 22 were valid according to the quality criteria of the measurements | |
| 13. In Table 4 , for C2 and C3a studies, 3 and 7 transects were measured, respectively. Elsewhere, the author stated that 12 to 16 transects should be applied. If so, will the small sampling size significantly enlarge the final error? Can you demonstrate it further? | **Clarification provided**

According to the text, the 12 to 16 transect are for measurements in refineries according to the referred normative. For farms, 5 to 6 transects are ideal because there is less variation. | **No modification** |
| 14. In Figure 8b caption,  the flux on the figure corresponded to 0.55 kg/h. However, in the main text line 425 said 'the SOF could measure concentration as low as 1 kg/h with an uncertainty of 21%'. But 0.55 kg/h is below such threshold. which statement is true?  What is the lowest detection limitation to use SOF measuring Nh3 sources? Shall it be further explored or is it proved by current study? | **Clarification provided**

Yes this is inconsistent. We added 0-5-1 kg/h in the text. | **Added line 23:**

(~0.5 - 1 kg h$^{-1}$) and |

---

## Author Response (AR2)

**Response to reviewers for manuscript**

Dear Senior Editor,

Thank you for the opportunity to revise our manuscript and for the feedback on these minor points. We implemented all the changes, and we believe that the manuscript has now improved its quality.

We look forward to your decision.

Best Regards,

Johan Mellqvist, Nathalia Thygesen Vechi, and co-authors

| Reviewer Comment | Author' Response | Revised Text – Line numbers refer to clean (without track changes) version of the revised paper. |
|---|---|---|
| Line 105: Consider removing one "ideally". Line 110: " introduce" should be "introducing". Maybe consider if this entire sentence is even necessary. | Suggestion implemented according to the editor suggestions | |
| Section 2.2.1 and 3.3: the mathematical symbol /alpha is used for two different quantities in the manuscript (relative wind angle in eq 1 and in eq. 10). Please rename it for one of these instances. | **Suggestion implemented** We change the α from equation 10 to an "r" of ratio. | Changes in Eq. 10 and 11 and on line 328. $$r = \frac{\log(U_2/U_1)}{\log(z_2/z_1)}$$ $$U_{(z)} = U_2\left(\frac{z}{z_2}\right)^r$$ Thereafter, the obtained r factor was … |
| Line 185: Ineris is an acronym. Please spell out the full name. Line 193. Add "Inc." to Fluxsense. Not everyone knows this is a company. | Suggestion implemented according to the editor suggestions | |

| | | |
|---|---|---|
| Line 199. Add "distance" after 250 – 900m (assuming this is what you meant) | | |
| Line 258: I agree with reviewer 1 that adding a sentence explaining the factor 1.96 would improve the manuscript. It is not well enough known for every reader to understand where this number is coming from. | **Suggestion implemented.**

An extra comment was added in the sentence. | Line 260: Therefore, it ($U_{cros}$) was calculated using absorption strength ($U_{abs-NH_3}$) (Kleiner et al., 2003), further divided by 1.96, **which is the coverage factor used for 95 % confidence interval**, as this error was considered a normal distribution (Eq. 5). |
| Line 489: replace "their" with "its" since you are referring to the "stickiness" of NH3 not of its emissions.
Line 507: Replace "very" with "vary"
Line 539: "instrument" should be "instruments" | **Suggestion implemented according to the editor suggestions** | |